# An atmospheric inversion over the city of Cape Town: sensitivity analyses

Alecia Nickless[1,2], Peter J. Rayner[3], Robert J. Scholes[4], Francois Engelbrecht[4], and Birgit Erni[2,5]

[1]Atmospheric Chemistry Research Group, School of Chemistry, University of Bristol, Bristol, BS8 1TS, UK
[2]Department of Statistical Sciences, University of Cape Town, Cape Town, 7701, South Africa
[3]School of Earth Sciences, University of Melbourne, Melbourne, VIC 3010, Australia
[4]Global Change Institute, University of the Witwatersrand, Johannesburg, 2050, South Africa
[5]The Centre for Statistics in Ecology, the Environment and Conservation, University of Cape Town, Cape Town, 7701, South Africa

**Correspondence:** Alecia Nickless alecia.nickless@bristol.ac.uk

**Abstract.** An atmospheric inversion was performed for the City of Cape Town for the period March 2012 to June 2013, making use of in situ measurements of $CO_2$ concentrations at temporary measurement sites located to the North East and South West of Cape Town. This paper presents results of sensitivity analyses which tested assumptions regarding the prior information and the uncertainty covariance matrices associated with the prior and with the observations. Alternative prior products were considered in the form of a carbon assessment analysis to provide biogenic fluxes and the ODIAC (Open-source Data Inventory for Anthropogenic $CO_2$ product) fossil fuel product. These were used in place of the reference inversion's biogenic fluxes from CABLE (Community Atmosphere Biosphere Land Exchange model) and fossil fuel emissions from a bespoke inventory analysis carried out specifically for the Cape Town inversion. Our results confirmed that the inversion solution was strongly dependent on the prior information, but by using independent alternative prior products to run multiple inversions, we were able to infer limits for the true domain flux. Where the reference inversion had aggregated prior flux estimates that were made more positive by the inversion – suggesting that CABLE was overestimating the amount of $CO_2$ biogenic uptake – the carbon assessment prior fluxes were made more negative by the inversion. As the posterior estimates were tending towards the same point, we could infer that the best estimate was located somewhere between these two posterior fluxes.

The inversion was shown to be sensitive to the spatial error correlation length in the biogenic fluxes – even a short correlation length – influencing the spatial distribution of the posterior fluxes, the size of the aggregated flux across the domain, and the uncertainty reduction achieved by the inversion. Taking advantage of expected spatial correlations in the fluxes is key to maximising the use of a limited observation network. Changes to the temporal correlations in the observation errors had very minor affects on the inversion.

The control vector in the original version consisted of separate day and night-time weekly fluxes for fossil fuel and biogenic fluxes over a four-week inversion period. When we considered solving for mean weekly fluxes over each four week period – i.e. assuming the flux remained constant over the month – larger changes to the prior fossil fuel and biogenic fluxes were possible, as well as further changes to the spatial distribution of the fluxes compared with the reference. The uncertainty reduction achieved in the estimation of the overall flux increased from 25.6% for the reference inversion to 47.2% for the mean weekly

flux inversion. This demonstrates that if flux components that change slowly can be solved for separately in the inversion, where these fluxes are assumed to be constant over long periods of time, the posterior estimates of these fluxes substantially benefit from the additional observational constraint.

In summary, estimates of Cape Town fluxes can be improved by using better and multiple prior information sources, particularly on biogenic fluxes. Fossil fuel and biogenic fluxes should be broken down into components, building in knowledge on spatial and temporal consistency in these components into the control vector and uncertainties specified for the sources for the inversion. This would allow the limited observations to provide maximum constraint on the flux estimates.

## 1 Introduction

Bayesian inverse modelling provides a top-down technique for verifying emissions and uptake of carbon dioxide ($CO_2$) from both natural and anthropogenic sources. It relies on accurate measurements of $CO_2$ concentrations at suitably located sites which can collect information about these sources at different spatial and temporal scales. The concentration measurements on their own are not sufficient to solve for the emission sources as there are many more sources of $CO_2$ than there are measurements of the concentrations. Therefore well-informed initial estimates of the biogenic and anthropogenic emissions are required, together with uncertainty estimates, which are used to regularise the problem. This technique is a useful tool for monitoring, reporting and verification (MRV) of $CO_2$ emissions from cities (Bellassen and Stephan, 2015; Wu et al., 2016; Lauvaux et al., 2016; Oda et al. , 2017a). While cities represent only 2% of the global land surface area, they are responsible for approximately 70% of anthropogenic greenhouse gas emissions (UN–Habitat, 2011; Seto et al., 2014), with annual urban $CO_2$ emissions averaging more than double the size of net terrestrial or ocean carbon sinks (Le Quéré et al., 2013).

Estimates of city-level $CO_2$ emissions are usually obtained using bottom-up techniques, which require knowledge of what activities produce $CO_2$ emissions and the fuel usage of these activities. These estimates are strongly dependent on accurate reporting, accurate and representative emission factors, and on assumptions regarding temporal or spatial disaggregation of these emissions (Andres et al., 2012). Ascertaining the uncertainty in these inventory-based estimates is not trivial, and these uncertainties increase as the spatio-temporal resolution of these estimates is increased (Turnbull et al., 2011; Andres et al., 2014).

Verifying the accuracy of inventory-based estimates of emissions has become essential (NRC, 2010). This requires transparency, quality and comparability of information, with narrow uncertainty estimates (Wu et al., 2016), but currently uncertainties associated with urban emissions far exceed emission reduction goals, and therefore verification remains challenging. The uncertainty is due to factors such as incomplete data, inconsistency in reporting between different institutions or facilities, fugitive emissions from point sources such as those caused by gas leaks, and methodology which is rarely checked against scientific standards and procedures (Hutyra et al., 2014). Recently several inverse modelling studies aimed at resolving $CO_2$ emissions have been conducted at the city-scale in Europe and North America (Strong et al., 2011; Duren and Miller, 2012; McKain et al., 2012; Brioude et al., 2013; Kort et al., 2013; Lauvaux et al., 2013; Bréon et al., 2015; Turnbull et al., 2015; Boon et al., 2016; Lauvaux et al., 2016; Oda et al. , 2017a), and more recently for the city of Cape Town (CT) in South Africa

(Nickless et al., 2018). South Africa is the single largest emitter of $CO_2$ on the continent of Africa, and the 13th largest emitter in the world (Boden et al., 2011). South African cities are home to 63% of the present population (Statistics South Africa, 2011), and by 2030 this is predicted to be 71%. Cape Town saw its population increase from 2,563,095 in 1996 to 3,740,026 in 2011, an increase of 46% (City of Cape Town, 2011).

Atmospheric inversions at the city-scale are limited by available $CO_2$ concentration observations – due to insufficient monitoring sites, but also a limited number of locations for suitable monitoring sites (Bréon et al., 2015). Atmospheric transport is complex in the urban environment and challenging for atmospheric transport models to resolve. This may result in large representation errors in the modelled concentrations at the measurements sites. To avoid these errors, a further reduction in the number of observations is often made, as observations are excluded based on when the models are likely to perform poorly

(Lauvaux et al., 2016; Staufer et al., 2016). The observed concentration data are as a result of aggregated fluxes from all sources of $CO_2$ along the path of the air flow. Sources refer to anything which may have a positive (i.e. emit) or negative (i.e. uptake) contribution to the overall $CO_2$ concentration. Even if biogenic fluxes are not necessarily of interest in the city-level inversion, they need to be accounted for in the model as these fluxes will induce changes to the observed $CO_2$ concentration.

Atmospheric monitoring sites targeting CT air masses were not available, therefore temporary measurement sites were
installed at Robben Island and Hangklip lighthouses, located to the North West and South East of the metropolis (Nickless et al., 2018). A fossil fuel emission inventory analysis was performed for the city which spatially and temporally disaggregated these fluxes to provide prior estimates of the fossil fuel fluxes, with uncertainty estimates determined by means of error propagation techniques (Nickless et al., 2015a). Net ecosystem exchange (NEE) fluxes from biogenic processes were obtained from the land atmosphere exchange model CABLE (Community Atmosphere Biosphere Land Exchange). Uncertainty estimates were
based on the estimates of net primary productivity (NPP). CABLE was dynamically coupled to the regional climate model CCAM (Conformal Cubic Atmospheric Model), which provided the climate inputs required to drive the Lagrangian particle dispersion model (LPDM). The Bayesian inversion framework included a control vector where fossil fuel and NEE fluxes were solved for separately.

One way that CT differs from the mega cities that previous inversions have targeted (Bréon et al., 2015; Staufer et al., 2016)
is through the high integration of natural areas around the city borders of CT (Nickless et al., 2018). Natural fluxes are an important contributor to the $CO_2$ budget of the region. For example, Table Mountain National Park is located directly adjacent to the city bowl and covers an area of $221\,\text{km}^2$. For this reason, the gradient method used by Bréon et al. (2015) and Staufer et al. (2016), which relies on the difference between pairs of measurement sites when the wind is blowing from one site, over the target region, to the second site, would not be appropriate given the locations of our two measurement sites. For the CT
case, if the air travelled between the two sites, it would pass directly over Table Mountain National Park, and therefore the gradient method would not have the desired effect of diminishing the impact of biogenic fluxes along the transect between the two sites. In addition, the wind fields showed that air did not travel in a straight path between our two sites (Nickless et al., 2018).

We adopted the approach usually used from regional inversions, where the inversion modelled the concentrations at the
measurement sites (Lauvaux et al., 2012). Instead of subtracting the background $CO_2$ concentration from the measurements,

which would have arrived from one of the domain boundaries, we solved for the concentrations at the boundary as an additional unknown, and therefore included these in the control vector, similar to the approach of Lauvaux et al. (2016). We kept tight constraints on these concentrations, and used the background measurements obtained from Cape Point, a Global Atmospheric Watch (GAW) background station, as prior estimates of these concentrations. We were able to do this as there are no large anthropogenic sources near the boundary of the domain. We showed in the reference inversion that the variation in the total $CO_2$ was largely driven by the variation in the NEE fluxes (Nickless et al., 2018).

Nickless et al. (2018) was a first attempt at estimating $CO_2$ fluxes at the high resolution of $1 \, \text{km}$ by $1 \, \text{km}$ over CT, solving for separate fossil fuel and biogenic sources. The inversion increased the domain emission of $CO_2$ from -83.5 kt per month to -19.8 kt. The inversion was able to reduce uncertainty of the total flux within a pixel by up to 97.7%, and was able to reduce the uncertainty in the total weekly flux over the whole domain by up to 50.5%. The largest innovation to a fossil fuel flux was applied to the pixel with the largest point-source fossil fuel flux over an oil-refinery. We found that the optimal solution for the posterior fluxes was one which made the overall flux in this pixel less positive by reducing the fossil fuel flux and by creating areas of more negative fluxes around this pixel. This indicated that either the prior fossil fuel flux was over-estimated, or the atmospheric transport model was not correctly indicating sensitivity of the measurement site to this flux. Compared with the fossil fuel emissions, relative innovations to the NEE fluxes were much larger, due to the large uncertainty assigned to these fluxes. The largest innovations were made to natural areas near the central business district (CBD) of CT, as well as to agricultural regions within the domain, particularly those close to the measurement sites.

Nickless et al. (2018) demonstrated the advantage of using the Bayesian inverse modelling approach to solve for disaggregated fluxes within each pixel when the ultimate goal was to solve for the aggregated flux within each pixel or within a region of interest. The inversion created negative covariances in the posterior uncertainty covariance matrix for those fluxes that were viewed simultaneously at the atmospheric measurement site. When we summed these fluxes, the effect of these negative covariances was to reduce the uncertainty of the aggregated flux – over and above the uncertainty reduction achieved by the inversion for the individual fluxes.

The specification of the uncertainty covariance matrices substantially influences the inversion result (Lauvaux et al., 2016). This paper investigates a series of adjustments to the inversion which impact on the uncertainty covariance matrix of the fluxes and the observation error covariance matrix. We considered sensitivity tests which halved and doubled the uncertainties of the individual sources, and investigated the impact of the uncertainty correlations in the inversion. We also manipulated the prior products, either by smoothing the products used in the reference inversion, or using alternative sources for the fossil fuel and biogenic prior fluxes and uncertainties.

Additionally we were interested in the composition of the control vector, also referred to as the state vector, which specifies the surface fluxes and domain boundary concentrations to be solved for by the inversion. The composition of this vector is determined by the size of the source pixels and the time length over which we assume the fluxes are homogeneous. This in turn impacts on the assigned uncertainty covariance matrix. For the reference inversion we carried out thirteen four-week inversions which solved for weekly fluxes from each of the $101 \times 101$ surface pixels. The weekly fluxes consisted of working week and weekend fossil fuel fluxes, and NEE fluxes for the full week; each separated into day and night fluxes. We tested

whether solving for an average weekly flux over the course of four weeks would achieve similar results compared with the reference inversion, which allowed the four weekly fluxes within a monthly inversion to differ. We also compared the reference inversion with the approach of carrying out separate inversions for each week. Each of these cases requires considerably less computational resources to perform an individual inversion. If either of these alternative control vectors provides sufficiently

similar results to the reference case, this would provide a more efficient means of conducting the inversion.

The purpose of this paper is to present the results of these sensitivity tests in comparison with the CT reference inversion presented in Nickless et al. (2018), with the aim of determining the best course of action to improve the ability to resolve fluxes for CT through the inversion method. Section 2 briefly introduces the Bayesian inversion framework used in the reference inversion (Nickless et al., 2018). This is followed by a description of the alternative prior information products and a presentation

of the details of the sensitivity analyses. A summary of the reference inversions and the results of the sensitivity analyses are provided in section 3, followed by discussion of these results in section 4, and a final concluding section.

## 2   Methods

### 2.1   Reference Inversion and Bayesian Inverse Modelling Framework

#### 2.1.1   Bayesian Inverse Modelling Approach

The Bayesian synthesis inversion method, as described by Tarantola (2005) and Enting (2002), was used to solve for the fluxes in this study. The observed concentration ($c$) at a measurement station results from contributions from the surface in the form of fluxes, from the domain boundaries, and from the initial concentration at the site. Concentrations at the measurement site can be modelled as:

$$c_{mod} = \mathbf{H}s \tag{1}$$

where $c_{mod}$ are the modelled concentrations and $s$ a vector of source fluxes or concentrations. $\mathbf{H}$ is the Jacobian matrix representing the first derivative of the modelled concentration at the observational site and dated with respect to the coefficients of the source components (Enting, 2002). It provides the sensitivity of each observation to each of the sources, where the sources can be fluxes or concentrations of $CO_2$. Estimates of the unknown sources can be obtained by minimising the following

cost-function with respect to $s$:

$$J(s) = \frac{1}{2} \left( (c_{mod} - c)^T \mathbf{C}_c^{-1} (c_{mod} - c) + (s - s_0)^T \mathbf{C}_{s_0}^{-1} (s - s_0) \right) \tag{2}$$

where $s$ is the control vector of unknown surface fluxes and boundary concentrations we wish to solve for, $s_0$ is the vector of

prior flux and boundary concentration estimates, $\mathbf{C}_c$ is the uncertainty covariance matrix of the observations, and $\mathbf{C}_{s_0}$ is the uncertainty covariance matrix of the fluxes and boundary concentrations (Tarantola, 2005).

Minimising this cost function leads to the following solution:

$$\mathbf{s} = \mathbf{s}_0 + \mathbf{C}_{s_0} \mathbf{H}^T \left( \mathbf{H} \mathbf{C}_{s_0} \mathbf{H}^T + \mathbf{C}_c \right)^{-1} (\mathbf{c} - \mathbf{H} \mathbf{s}_0) \tag{3}$$

with posterior covariance matrix:

$$\mathbf{C}_s \;\; = \;\; \left( \mathbf{H}^T \mathbf{C}_c^{-1} \mathbf{H} + \mathbf{C}_{s_0}^{-1} \right)^{-1} \tag{4}$$
$$= \;\; \mathbf{C}_{s_0} - \mathbf{C}_{s_0} \mathbf{H}^T \left( \mathbf{H} \mathbf{C}_{s_0} \mathbf{H}^T + \mathbf{C}_c \right)^{-1} \mathbf{H} \mathbf{C}_{s_0}. \tag{5}$$

### 2.1.2 Control Vector - $s$

The total $CO_2$ flux from a single surface pixel can be thought of as being made up of the following individual fluxes:

$$\mathbf{s}_{sf;\,i} = \mathbf{s}_{ff\ week\ day;\,i} + \mathbf{s}_{ff\ week\ night;\,i} + \mathbf{s}_{ff\ weekend\ day;\,i} + \mathbf{s}_{ff\ weekend\ night;\,i} + \mathbf{s}_{NEE\ day;\,i} + \mathbf{s}_{NEE\ night;\,i} \tag{6}$$

where $\mathbf{s}_{sf;\,i}$ is the total weekly surface flux from the $i^{th}$ pixel, $\mathbf{s}_{ff\ week\ day;\,i}$ is the total fossil fuel flux during the working week day, $\mathbf{s}_{ff\ week\ night;\,i}$ is the total night-time fossil fuel flux during the working week, $\mathbf{s}_{ff\ weekend\ day;\,i}$ is the total weekend

daytime fossil fuel flux, $\mathbf{s}_{ff\ weekend\ night;\,i}$ is the total weekend night-time fossil fuel flux, and $\mathbf{s}_{NEE\ day;\,i}$ and $\mathbf{s}_{NEE\ night;\,i}$ are the total day and night-time biogenic fluxes for the full week from the $i^{th}$ spatio-temporal pixel. The reference inversion solved for each of these separate fluxes for each week. There are $101 \times 101 = 10{,}201$ surface pixels. Over the 16 month period from March 2012 to June 2013, separate monthly inversions were carried out for all months with sufficient valid concentration observations; a total of 13 inversions. Each monthly inversion solved for four weekly fluxes. Therefore a monthly inversion

solves for $10{,}201 \times 6 \times 4 = 244{,}824$ surface fluxes.

The mean day and night-time concentrations at each of the four domain boundaries for each week are included in the control vector. The inversion solved for $4 \times 2 \times 4 = 32$ boundary concentrations (4 boundaries, day/night, 4 weeks). We solved for weekly concentrations at the boundaries as we expected these concentrations to show small changes on synoptic time scales, particularly inflow from the ocean boundaries. We avoided solving for too short a period so that the percentile filtering

technique (see section 2.1.8) would never discard all measurements for a period. The maximum standard deviation in the hourly background $CO_2$ concentrations for a week was 0.8 ppm.

### 2.1.3 Concentration Measurements - $c$

The reference inversion made use of two $CO_2$ monitoring sites that were established at Robben Island and Hangklip lighthouses. Each site was equipped with a Picarro Cavity Ring-down Spectroscopy (CRDS) (Picarro G2301) instrument. Sufficient data for 13 of the 16 months were available to perform monthly inversions. The Robben Island site viewed predominantly air influenced by the Cape Town city bowl whereas Hangklip viewed air influenced by biogenic fluxes from nearby fynbos vegetation and agricultural areas. Details about these measurement sites are provided in Nickless et al. (2018). Rigorous calibration was performed on a regular basis, ensuring that these sites measured on the same scale as the Cape Point background site, which is calibrated to the WMO-X2007 scale. The high frequency observations were processed into hourly concentrations which provided the observed data for the inversion.

### 2.1.4 System Meteorology

CCAM is a variable-resolution global atmospheric model developed by the Commonwealth Scientific and Industrial Research Organisation (CSIRO) (McGregor, 1996; McGregor and Dix, 2001; McGregor, 2005a, b; McGregor and Dix, 2008), and has been validated over South Africa (Engelbrecht et al., 2009; Roux, 2009; Engelbrecht et al., 2011, 2013, 2015). Full details are provided in Nickless et al. (2018). CCAM was applied in stretched-grid mode to function as a regional climate model. A multiple-nudging approach was followed to downscale the 250 km resolution National Centres for Environmental Prediction (NCEP) reanalysis data (Kalnay et al., 1996) to a resolution of 60 km over southern Africa, 8 km over the south western Cape and subsequently to a 1 km resolution over the study area. The model produced hourly estimates on a 1 km × 1 km spatial grid, which extended from $34.5°$ to $33.5°$ south and from $18.2°$ to $19.2°$ east.

### 2.1.5 Jacobian Matrix - H

The Jacobian matrix, $\mathbf{H}$, provides the sensitivities of the concentrations observed at the receptor sites to the surface fluxes and boundary inflows. To generate this matrix in our application the particle counts were processed from a Lagrangian particle dispersion model (LPDM) run in backward mode (Uliasz, 1994). The LPDM was driven by hourly three-dimensional fields of mean winds ($u$, $v$, $w$), potential temperature and turbulent kinetic energy (TKE), which were obtained from the CCAM model. LPDM simulates atmospheric transport by releasing particles from the observational sites and tracking these particles backward in time. These particle counts were used to derive the elements of the Jacobian matrix $\mathbf{H}$ as originally described by Seibert and Frank (2004) and subsequently used in several inversion studies (Lauvaux et al., 2012; Wu et al., 2013; Ziehn et al., 2014; Nickless et al., 2015b; Lauvaux et al., 2016; Oda et al. , 2017a; Nickless et al., 2018).

Previously we modified the approach of Seibert and Frank (2004) to use particle counts – as produced by our LPDM – instead of mass concentrations which were output by the atmospheric transport model FLEXPART in their study (Ziehn et al., 2014) . The elements of the matrix $\mathbf{H}$ corresponding to the surface fluxes in $s$ were calculated as follows:

$$\frac{\partial \boldsymbol{c}_{sf}^-}{\partial \boldsymbol{s}_{in}} = \frac{\Delta T \mathrm{g}}{\Delta P} \overline{\left(\frac{N_{in}}{N_{tot}}\right)} \frac{44}{12} \times 10^3, \tag{7}$$

where $\boldsymbol{c}_{sf}^-$ is a volume mixing ratio (receptor) expressed in ppm and $\boldsymbol{s}_{in}$ is a mass flux density (source), $N_{in}$ the number of particles in the receptor surface grid from source pixel $i$ released at time interval $n$, $\Delta T$ is the length of the time interval, $\Delta P$ is the pressure difference in the surface layer, g is the acceleration due to gravity, and $N_{tot}$ the total number of particles released
during a given time interval.

The spatial resolution of the surface flux grid boxes was set to be the same as that of the high-resolution subregion of the atmospheric transport model, resulting in a gridded domain consisting of $101 \times 101$ grid boxes (a resolution of $1\,\mathrm{km} \times 1\,\mathrm{km}$). The units of the surface fluxes are given in kg $CO_2$ m$^{-2}$ week$^{-1}$ and are transformed through $\mathbf{H}$ into contributions to the concentration at the measurement site in units of ppm. To solve for the concentrations at the boundary Ziehn et al. (2014)
showed that the Jacobian can be calculated as:

$$\frac{\partial \bar{\boldsymbol{c}}_b}{\partial \boldsymbol{s}_B} = \frac{N_B}{N_{tot}} \tag{8}$$

where $\boldsymbol{s}_B$ are the concentrations at the domain boundary, $\bar{\boldsymbol{c}}_b$ is the volume mixing ratios, $N_B$ is the number of particles from
the domain boundary, $B$, and $N_{tot}$ the total number of particles viewed at the receptor site from any of the domain boundaries. The contribution to the observed concentration at the receptor site can be written as:

$$\boldsymbol{c}_b = \mathbf{H}_B \boldsymbol{s}_B \tag{9}$$

where $\mathbf{H}_B$ is the Jacobian with respect to the domain boundary concentrations, $\boldsymbol{s}_B$ are the domain boundary concentrations
and $\boldsymbol{c}_b$ the contributions from the boundary to the observed concentration at the measurement site in units of ppm. The row elements of $\mathbf{H}_B$ sum to one. Therefore the elements of $\boldsymbol{c}_b$ represent a weighted average of the concentrations at the domain boundaries, and provide a basis concentration to which the contributions from the surface fluxes are added. Each inversion solves for weekly domain boundary concentrations at the northern, eastern, southern and western borders of the inversion domain box, separated by day and night.

**2.1.6   Inventory of Anthropogenic Emissions**

The inventory analysis carried out for CT subdivided the anthropogenic emissions into road transport, airport and harbour, residential lighting and heating, and industrial point source emissions (Nickless et al., 2015a). Road transport emissions were derived from modelled values of vehicle kilometres for each section of the road network, based on observed vehicle count data. The vehicle kilometres were scaled for each hour of the day, and separated into week days and weekend days, leading

to distinctive vehicle emissions for the week / weekend and day / night periods. Airport emissions were derived from landing and takeoff cycles, as reported by Airports Company South Africa for each month. The IPCC average emission factors for domestic and international fleets (IPCC, 2000) were used to convert the airport activity data into emissions of $CO_2$. Harbour emissions were derived from gross tonnage of vessels which docked at CT port during each month published by the South

African Ports Authority, and emissions derived as described in DEFRA (2010). Residential emissions for lighting and heating were derived from population count data obtained for each of the municipal wards in 2011 (Statistics South Africa, 2011). The South African government reports on the fuel used for domestic heating and lighting (South African Department of Energy, 2009). This was divided between the total population, and then allocated pro rata to each ward. It was assumed that 75% of the annual energy consumed was used for heating, 20% for cooking and 5% for lighting. The majority – 75% – of the

emissions for heating were allocated to the winter months. CT provided monthly fuel use for the largest industrial emitters. These were converted directly into $CO_2$ emissions by multiplying the fuel amount with the DEFRA greenhouse gas emission factors (DEFRA, 2013a). The fuel types that were considered included heavy fuel oil, coal, diesel, paraffin and fuel gas, which was divided into liquid petroleum gas and refinery fuel gas.

    Based on this inventory analysis, the percentage contribution of industrial point sources to the total fossil fuel emission for CT

was 12.0%, 34.6% from vehicle road transport, 51.0% from the residential sector, and 2.4% from airport and harbour transport. Residential emissions are a large contributor to the fossil fuel emission budget as well as one of the largest contributors to the uncertainties in the fossil fuel flux. This is due to the dependency that many people living in CT have on raw fossil fuel burning for heating and lighting. Emissions from power stations are a small component of the total fossil fuel flux from CT as the bulk of the direct emissions from power stations occur elsewhere in the country.

The total fossil fuel $CO_2$ emissions for the domain were within the range of $CO_2$ emissions reported in the EDGAR (Emission Database for Global Atmospheric Research) (v4.2) database (Nickless et al., 2015a). EDGAR is a global product on a $0.1° \times 0.1°$ grid, which provides the total anthropogenic emissions of $CO_2$ as estimated from proxy data such as population counts and information on the road transport network (Janssens-Maenhout et al., 2012). The total emissions from the inventory for 2012 were 22% higher than the EDGAR emissions reported for 2010. The emissions in the inventory tended to be con-

centrated over specific sources, such as over an oil-refinery or along the road network, whereas the EDGAR emissions were smoothed over the city region.

### 2.1.7   Biogenic Emissions

CCAM was dynamically coupled to the land surface model CABLE (Kowalczyk et al., 2006), which allows for feedbacks between land surface and climate processes, such as leaf area feedback on maximal canopy conductance and latent heat fluxes

(Zhang et al., 2013). This also has the consequence that the spatial resolution of the biogenic fluxes were at the same spatial resolution of $1\,\mathrm{km} \times 1\,\mathrm{km}$ as for the transport model. The model produces hourly estimates of net ecosystem exchange (NEE), which were aggregated into weekly (day and night) flux estimates in units of $\mathrm{kg\ CO_2\ m^{-2}\ week^{-1}}$, and used as the prior estimate of biogenic fluxes over the land surface.

The natural areas within the target domain of the inversion are dominated by the fynbos biome. This is a biodiverse biome, with many endemic species, and covers a relatively small area in South Africa, but a large proportion of the area within the domain of the inversion. The fynbos biome is poorly represented by dynamic vegetation models (Moncrieff et al., 2015), and its ability to simulate biogenic fluxes in the fynbos region is largely untested. CABLE was selected as the land atmosphere exchange model to couple with CCAM due to its development for regions in Australia which are similar to the savanna biome in South Africa. In addition to the natural vegetation, a large agricultural sector is within the proximity of CT, particularly vineyards and fruit orchards. The CT region experiences a Mediterranean climate with winter rainfall, with hot and dry summers and mild and wet winters. Significant NEE fluxes take place during both winter and summer periods, as biogenic activity in this region is limited by the amount of water availability, whereas temperatures are usually sufficiently high for plant production and respiration. The $CO_2$ fluxes over the ocean were obtained from a study which characterised the seasonal cycle of air-sea fluxes of $CO_2$ in the southern Benguela upwelling system off the South African west coast (Gregor and Monteiro, 2013).

### 2.1.8 Domain Boundary Concentrations

The presence of the Cape Point GAW station provided a source of background $CO_2$ concentrations for the inversion. The Cape Point station is located approximately $60\,\mathrm{km}$ south of CT within a nature reserve, situated on the southern-most tip of the Cape Peninsula at a latitude of $34°21'12.0''$ south and longitude of $18°29'25.2''$ east. The inlet is located on top of the $30\,\mathrm{m}$ measurement tower mounted on a cliff $230\,\mathrm{m}$ above sea level. The station observes background measurements of $CO_2$ when observing maritime air advected directly from the south-western Atlantic Ocean - an extensive region stretching from $20°$ (sub-equatorial) to $80°$ south (Antarctic region) (Brunke et al., 2004). Therefore, maritime measurements at Cape Point from the Southern Ocean are well representative of the background $CO_2$ signal influencing the Cape Peninsula, which are the concentrations expected at the boundary of the inversion domain. The background signal at Cape Point, represented by a subset of the measurements obtained from a percentile filtering technique (Brunke et al., 2004), was used as the prior estimate of the concentrations at each of the four domain boundaries. The percentile filtering technique removes data influenced by the continent or anthropogenic emissions. When applied to the Cape Point $CO_2$ measurements, approximately 75% of the data are selected. The percentile-filtering technique has been shown to compare well with the more robust method of using contemporaneous radon ($^{222}Rn$) measurements to differentiate between marine and continental air (Brunke et al., 2004).

The Cape Point measurements of the background $CO_2$ levels meant that we were not dependent on the atmospheric transport model to produce estimates of $CO_2$ concentrations at the domain boundary, which are prone to large errors (Lauvaux et al., 2016). The mean weekly background concentrations, separate for day and night, were determined from the percentile filtered measurements at the site, and were used as the prior domain boundary concentrations for each of the four cardinal directions. The prior uncertainty assigned to the boundary concentrations was set at the standard deviation of the measured hourly concentrations for that period, which resulted in a tight constraint on the prior background $CO_2$ concentrations. Large adjustments by the inversion to the domain boundary concentrations were not expected, including the terrestrial boundaries. The standard deviation in the hourly background $CO_2$ concentrations ranged between 0.32 and 0.90 ppm, with a mean of 0.62 ppm.

The boundaries of the domain were deliberately set to be far from the measurement sites so that contributions to the $CO_2$ concentration at a measurement site were dominated by the surface fluxes within the domain, rather than by the domain boundary concentrations.

### 2.1.9 Prior Uncertainty Covariance Matrix - $C_{s_0}$

Error propagation techniques, as described in Nickless et al. (2015a) and Nickless et al. (2018), were used to estimate the relative uncertainties for each of the sector specific fossil fuel estimates. The relative uncertainties were scaled by a value of 2 in order to ensure that the elements of the covariance matrix were statistically consistent with the assumptions of the inversion (Tarantola, 2005). The resulting uncertainty estimates (expressed as standard deviations) ranged between 6.7% to 71.7% of the prior fossil fuel emission estimate, with a median percentage of 34.9% to 38.4% depending on the month. These values

were more conservative compared with uncertainties of Bréon et al. (2015) for the AirParif inventory, which were set at 20% throughout. Since we solved for weekly, rather than daily fluxes, we used a strong assumption that fossil fuel fluxes within the same week were homogeneous over this time. To allow the inversion to react to local conditions within a given week, no temporal uncertainty correlation was assumed between weekly fluxes. Since fossil fuel emissions were expected to be localised in space, we also assumed no spatial uncertainty correlation between fossil fuel fluxes.

The uncertainty in the biogenic prior fluxes was set at the absolute value of the net primary productivity (NPP) as produced by CABLE. Therefore, the uncertainties assigned to the NEE estimates were large, but there is a great deal of uncertainty in both the productivity and respiration fluxes contributing to the NEE flux (Wang et al., 2011). The estimates of NEE are strongly dependent on the assumed model forms selected for different processes in the CABLE model. For example, the model forms used for the soil temperature-respiration function and the soil moisture-respiration function have large impacts

on the NEE estimates, with resulting NEE estimates differing by over 100% compared with eddy-covariance measurements (Exbrayat et al., 2013). The approach of assigning either the productivity or respiration component of NEE as the uncertainty has been used by Chevallier et al. (2010). We wished to avoid assigning fixed proportional uncertainty to the NEE estimates as, particularly for semi-arid regions, small NEE fluxes could occur as a result of both large productivity and respiration fluxes. Proportional uncertainties would lead to unrealistically low estimates of the uncertainty in NEE fluxes. This is different to the

approach used by Bréon et al. (2015), where an uncertainty level of 70% was assigned to biogenic fluxes, but in their case absolute NEE estimates were usually large in summer and expected to be small in winter. For the ocean fluxes, the standard deviations in the daily $CO_2$ fluxes from Gregor and Monteiro (2013) were assigned as the uncertainties.

     To estimate spatial uncertainty covariances in the NEE fluxes, we assumed an isotropic Balgovind correlation model as used in Wu et al. (2013). The off-diagonal covariance elements for $s_{NEE;i}$ and $s_{NEE;j}$ were calculated as:

$$C_{s_{0;NEE}}(s_{NEE;i}, s_{NEE;j}) = \sqrt{C_{s_{0;NEE}}(s_{NEE;i})} \sqrt{C_{s_{0;NEE}}(s_{NEE;j})} (1 + \frac{h}{L}) exp(-\frac{h}{L}) \tag{10}$$

where $s_{NEE;i}$ and $s_{NEE;j}$ are NEE fluxes in pixels $i$ and $j$, $C_{s_{0;NEE}}(s_{NEE;i})$ and $C_{s_{0;NEE}}(s_{NEE;j})$ the corresponding variances

in the NEE flux uncertainties in pixels $i$ and $j$, the characteristic correlation length $L$ was assumed to be $1\,\mathrm{km}$, and $h$ is the spatial distance between the centres of pixels $i$ and $j$.

### 2.1.10  Uncertainty Covariance Matrix of the Observations - $\mathbf{C_c}$

The observation uncertainties represented in $\mathbf{C_c}$ contain both the measurement error and the error associated with modelling the concentrations. We assigned a minimum uncertainty variance of $4\,\mathrm{ppm}^2$ for daytime observations and $16\,\mathrm{ppm}^2$ for night-time observations. These values were assigned as baseline (i.e. minimum) errors, and accounted for measurement errors, atmospheric transport modelling errors, aggregation errors and representation errors. These minimum errors are smaller than those for city-scale inversions conducted in the Northern Hemisphere. We justify the use of these values in our application since CT is a smaller city compared with the cities considered in the megacity applications, such as Paris and Indianapolis. Measurements of background $CO_2$ in the Southern Hemisphere have smaller variability compared with measurements in the Northern Hemisphere. For example, for the years 2012 to 2013 the standard deviation between the monthly $CO_2$ means for Mauna Loa GAW station in the Northern Hemisphere was $2.3\,\mathrm{ppm}$ (Tans and Keeling , 2016), whereas for the same time period at Cape Point the standard deviation between the monthly means was $1.6\,\mathrm{ppm}$.

We added additional error estimates to these minimum observation errors. We assumed errors in modelled $CO_2$ concentrations due to the transport model would be larger when the wind speed was lower (Bréon et al., 2015), and this would be compounded at night when the planetary boundary layer height was shallower and more stable (Feng et al., 2016). Additional error ranging between 0 and $1\,\mathrm{ppm}^2$ was added to the daytime uncertainty variance of $4\,\mathrm{ppm}^2$, linearly scaled depending on the wind speed, with $0\,\mathrm{ppm}^2$ added when wind speeds were high ($20\,\mathrm{m\,s}^{-1}$ or higher) and $1\,\mathrm{ppm}^2$ when the wind speed was close to zero. At night the additional uncertainty ranged between 0 and $16\,\mathrm{ppm}^2$. We also accounted for the standard deviation of the measured $CO_2$ concentrations during each hour. We assumed that variability within the instantaneous measurements at the site during an hour would be associated with larger errors in the atmospheric transport model. The variance of the observed instantaneous $CO_2$ concentrations within an hour was added to the overall uncertainty. Therefore each hour had a customised observation error dependent on the prevailing conditions at the measurement site. Therefore the total observation uncertainty variance for hour $k$ is given as:

$$C_c(k,k) = C_{c;base}{}^2 + C_{c;wind}{}^2 + C_{c;obs}{}^2 \tag{11}$$

where $C_{c;base}$ is the baseline observation error of $2\,\mathrm{ppm}$ during the day and $4\,\mathrm{ppm}$ during the night, $C_{c;wind}$ is the additional error due to the wind speed conditions which ranged between 0 and 1, and $C_{c;obs}$ is the standard deviation of the observed concentrations within that hour. The final observation uncertainties reached up to $15\,\mathrm{ppm}$ at night, reducing the weight of these measurements in the estimation of the prior fluxes.

The off-diagonal elements of $\mathbf{C_c}$ were calculated, based on the Balgovind correlation model as used in Wu et al. (2013), as:

$$C_c(c_i, c_j) = \sqrt{C_c(c_i)}\sqrt{C_c(c_j)}(1 + \frac{h}{L})exp(-\frac{h}{L}) \tag{12}$$

where $c_i$ and $c_j$ are the average concentrations during hours $i$ and $j$, $C_c(c_i)$ and $C_c(c_j)$ the corresponding error variances for

the concentrations in hours $i$ and $j$, the characteristic correlation length $L$ was assumed to be $1\,\mathrm{hour}$, and $h$ is the length in time between observations $i$ and $j$. The impact of this, albeit short, correlation length was assessed in a sensitivity tests discussed in the next section. No consensus has yet been reached on how these observation uncertainty correlations should be treated in city-scale inversions (Lauvaux et al., 2016).

## 2.1.11 Model Assessment

In order to assess the appropriateness of the uncertainty covariance matrices $\mathbf{C}_{\boldsymbol{c}}$ and $\mathbf{C}_{\boldsymbol{s}_0}$, the $\chi^2$ statistic, as described in Tarantola (2005), was calculated as:

$$\chi^2 = (\mathbf{H}\boldsymbol{s}_0 - \boldsymbol{c})^T (\mathbf{H}\mathbf{C}_{\boldsymbol{s}_0}\mathbf{H}^T + \mathbf{C}_{\boldsymbol{c}})^{-1} (\mathbf{H}\boldsymbol{s}_0 - \boldsymbol{c}) \tag{13}$$

with degrees of freedom equal to $\nu$, the dimension of the data space – in this case the length of observations in the inversion.

The squared residuals from the inversion (squared differences between observed and modelled concentrations) should follow the $\chi^2$ distribution with degrees of freedom equal to the number of observations (Michalak et al., 2005; Tarantola, 2005). The expected value of $\chi^2/\nu$ is one. Values lower than one indicate that the uncertainty is too large, and values greater than one indicate that the uncertainty prescribed is lower than it should be. The error in the assignment of the uncertainty could be in either $\mathbf{C}_{\boldsymbol{c}}$ or $\mathbf{C}_{\boldsymbol{s}_0}$ (or both). In order to ensure the suitability of $\mathbf{C}_{\boldsymbol{s}_0}$, the prior uncertainty variances were multiplied by a factor of two. This ensured that the $\chi^2/\nu$ statistic was close to a value of one for almost all months of the inversion. These details are provided in Nickless et al. (2018). Due to the length of time it takes to run a single inversion, we did not calculate an individual scaling parameter for each month.

## 2.2 Sensitivity Tests

### 2.2.1 Alternative biogenic flux product

As part of a project assessing the carbon sinks of South Africa (DEA, 2015), monthly $1\,\mathrm{km} \times 1\,\mathrm{km}$ estimates of terrestrial carbon stocks and fluxes were produced (Scholes et al., 2013). To estimate these fluxes, a distinction was made between carbon stocks in natural to semi-natural areas and those on transformed land, such as annually-cropped cultivated land, plantation forests, and urban areas (which was based on the IPCC 2006 value for closed urban forests). As a sensitivity test, the NEE and NPP from CABLE estimates used for the biogenic flux priors and their uncertainties were replaced with NEE and NPP from the carbon assessment product and the inversion rerun with these priors (inversion S1).

To estimate gross primary productivity (GPP), ten years (2001 to 2010) of monthly climatologies (temperature, rainfall, relative humidity) and satellite products for photosynthetically active radiation (PAR) and fraction of absorbed photosynthetically active radiation (FAPAR) were assimilated. Autotrophic respiration (Ra) was calculated based on the inputs for temperature, above-ground biomass, below-ground biomass and FAPAR. NPP could then be calculated as NPP = GPP - Ra. The hetrotrophic component (Rh) of Ecosystem respiration (Re) was based on estimates of soil organic carbon stocks and above-ground litter.

The basic calculation to obtain NEE was NEE = GPP - Re, and additional losses of $CO_2$ through biomass burning, and export and import fluxes from harvest and trade-related activities were accounted for.

To disaggregate the monthly products into day and night fluxes, it was assumed that all GPP took place during the day, and that half of Re occurred during the day and half at night. Therefore the weekly NEE and NPP estimates used for the prior information in the inversion were based on the GPP and respiration products from the assessment. The carbon assessment estimated the GPP flux for the year in the fynbos biome to be $521 \, \mathrm{g} \, CO_2 \, \mathrm{m}^{-2} \mathrm{year}^{-1}$ with a standard deviation of $492 \, \mathrm{g} \, CO_2$ $\mathrm{m}^{-2} \mathrm{year}^{-1}$ across pixels with $1 \, \mathrm{km}^2$ resolution. Therefore, as for the CABLE estimates used in the reference inversion, we assign uncertainties to the prior NEE estimates equal to the NPP estimate. A map of the prior daytime NEE fluxes in May 2012 from the CABLE and carbon assessment products is provided in Figure 1. The prior estimates from the two products are deliberately plotted on the same scale to illustrate how much more homogeneous the carbon assessment estimates are compared with those from CABLE. For the CABLE product, regions with the highest productivity are associated with the largest uncertainties.

The homogeneity of the biogenic $CO_2$ fluxes across the domain from the carbon assessment product can be explained by the products used as inputs for the estimation of the carbon stock components, such as FAPAR. These products would not be expected to differ considerably from pixel to pixel in this domain. CABLE predicts greater $CO_2$ uptake. The average $CO_2$ flux over the course of the study period and across the domain, was $-41 \, \mathrm{g} \, CO_2 \, \mathrm{m}^{-2} \mathrm{week}^{-1}$ according to the carbon assessment and $-172 \, \mathrm{g} \, CO_2 \, \mathrm{m}^{-2} \mathrm{week}^{-1}$ according to CABLE. Both these estimates can be considered small. The true flux is likely to be highly variable but close to carbon neutral over a long period of time (several years).

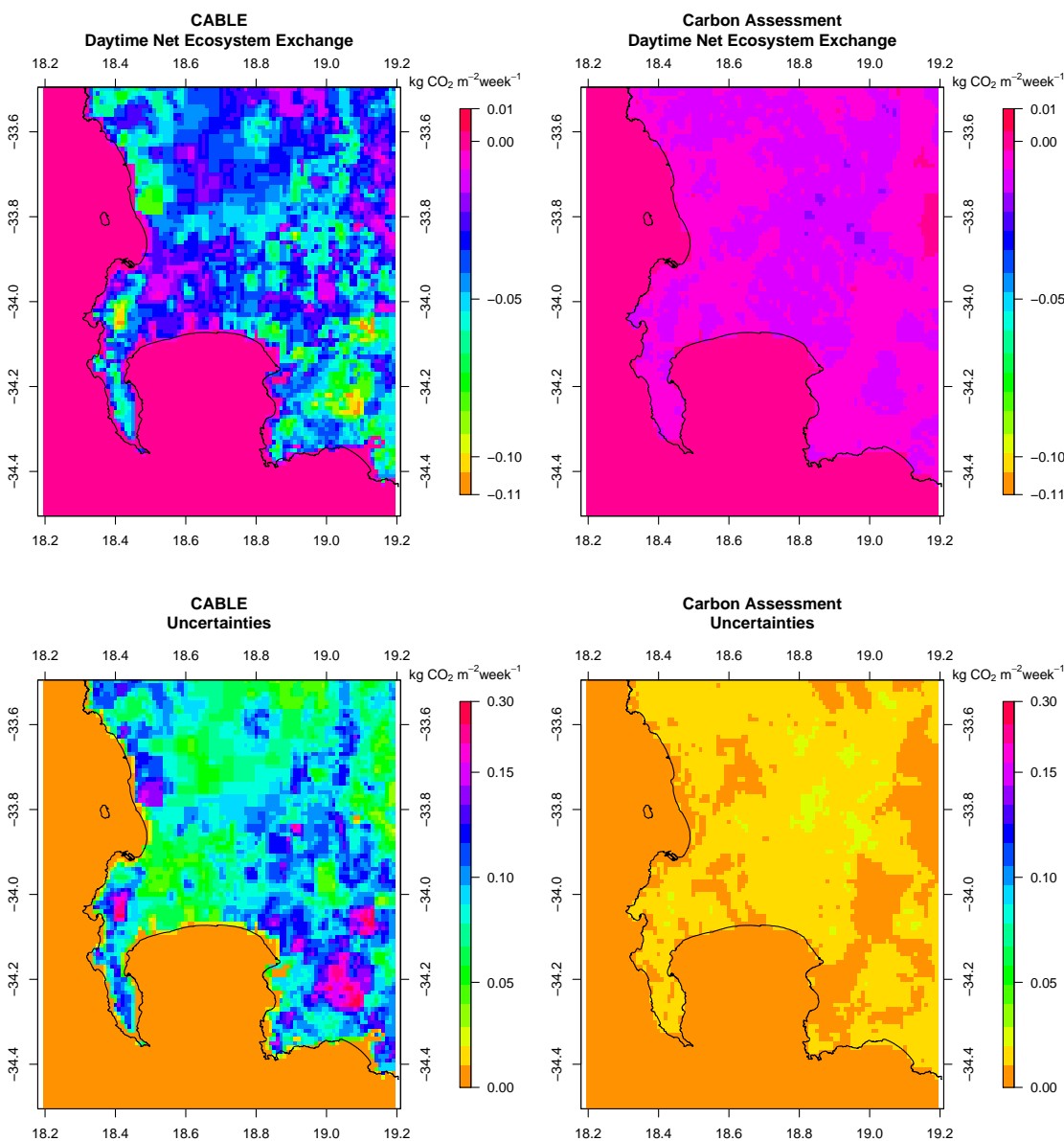

**Figure 1.** Spatial distribution of the prior daytime NEE fluxes produced by CABLE (top left) and the carbon assessment product (top right) for May 2012, as well as the uncertainty estimates assigned to these fluxes (bottom row). The scale for the prior NEE estimates is set between -0.11 and 0.01 $\mathrm{kg\ CO_2\ m^{-2}week^{-1}}$, whereas the scale for the uncertainties, determined from NPP estimates, ranges between 0.00 and 0.30 $\mathrm{kg\ CO_2\ m^{-2}week^{-1}}$.

### 2.2.2 Alternative fossil fuel emissions product

As an alternative to the inventory analysis of the fossil fuel fluxes, we used current estimates of anthropogenic fossil fuel emissions from the $1\,km \times 1\,km$ ODIAC product for the years 2012 and 2013 (ODIAC2017) (Oda and Maksyutov, 2011; Lauvaux et al., 2016; Oda et al. , 2017a, b) (inversion S2). The product provides monthly emissions of $CO_2$ in $kt$ of carbon.

The original ODIAC product (Oda and Maksyutov, 2011) made use of global energy consumption statistics and distributed the emissions from these activities based on known point source emitters, such as power plants, and on a global nightlight distribution satellite product. Emissions from point sources, such as those from power plants, were estimated separately from the diffuse emissions, for example those due to transport. These emissions were disaggregated onto to a $1\,km \times 1\,km$ grid. The updated product has further disaggregated the diffuse emissions to a $30\,m \times 30\,m$ grid by making use of global road

network data, a satellite product on surface imperviousness, and population census data (Oda et al. , 2017a, b). This $30\,m \times 30\,m$ diffuse emission product together with the point source emission product were aggregated back up to the $1\,km \times 1\,km$ grid. ODIAC has been shown to give comparable flux estimates when used in an inversion as a prior product in place of the ultra high resolution inventory product Hestia (Gurney et al., 2012), carried out for Indianapolis, IN (Oda et al. , 2017a).

The ODIAC monthly estimates were re-scaled according to the day of the week and to the hour of day using scaling factors

for South Africa as estimated by Nassar et al. (2013). These estimates were re-aggregated into day and night working week and weekend fossil fuel fluxes in units of $kg\ CO_2\ m^{-2}\ week^{-1}$. These estimates for the fossil fuel fluxes were used as prior estimates for the inversion in place of the inventory-based estimates used for the reference inversion. The daytime fossil fuel fluxes produced by the inventory analysis and the ODIAC product are provided in Figure 2.

The ODIAC product gave similar fossil fuel fluxes over pixels in the CBD area compared with the inventory estimates. The

inventory estimates were concentrated over the road network, point sources, and areas of high population density, whereas the ODIAC product dispersed emissions over the domain, with an area of high concentration over the CT metropolitan area and decreasing emissions away from this region. The average fossil fuel flux for the domain over the study period was $134\,g\ CO_2$ $m^{-2}week^{-1}$ according to the inventory and $274\,g\ CO_2\ m^{-2}week^{-1}$ according to the ODIAC product.

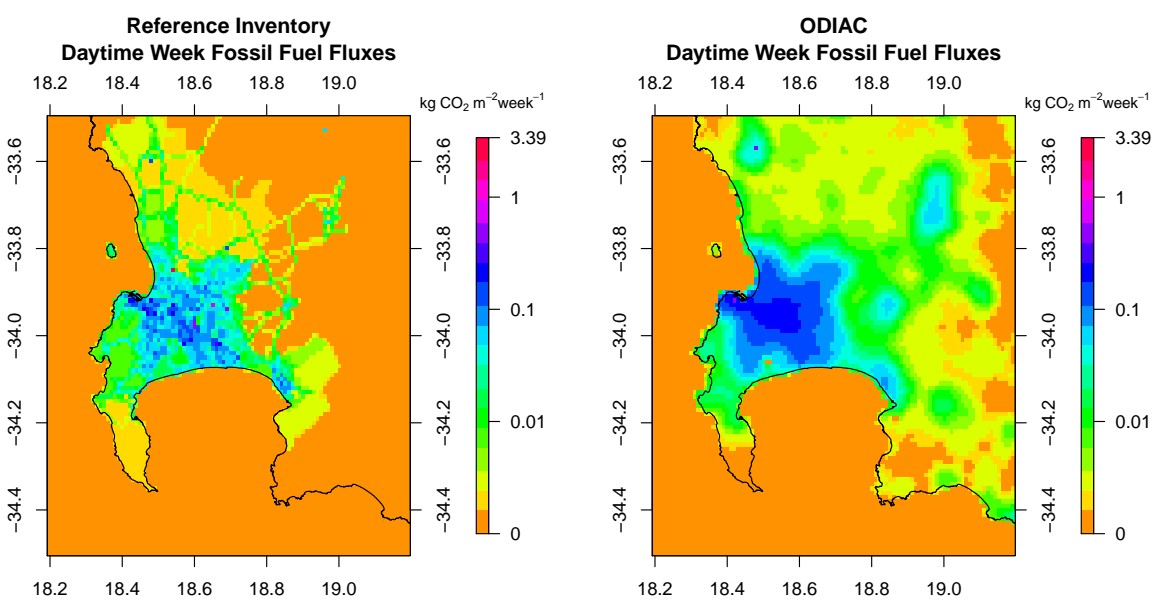

**Figure 2.** Spatial distribution of the prior fossil fuel fluxes produced from the Cape Town inventory analysis (left) and the ODIAC fossil fuel product (right) in May 2012. The prior estimates are plotted on a log scale which ranges between 0.00 and 3.39 $\mathrm{kg\ CO_2\ m^{-2} week^{-1}}$.

### 2.2.3 Alternative covariance structures

The specification of the prior uncertainty covariance structures has been shown to have a substantial impact on the pixel-level flux estimates, the total flux estimate for the domain, and on the spatial distribution of the fluxes (Wu et al., 2013; Lauvaux et al., 2016). For example, in the Indianapolis inversion, assuming correlation lengths of 4 or 12 km in the prior uncertainty covariance matrix of the fluxes resulted in total flux estimates for the city that were 17 and 25% larger than the total flux estimate assuming no correlation (Lauvaux et al., 2016). The effect of changing the correlation length had a larger impact on the total flux estimate than changing the prior emission product from Hestia to ODIAC.

To assess the sensitivity of the posterior flux estimates, their uncertainties, and their distribution in space to the specification of the uncertainty correlations, we ran inversions where the non-zero off-diagonal elements of $\mathbf{C}_{s_0}$ and $\mathbf{C}_c$ in the reference inversion were systematically set to zero. We considered an inversion which assumed no temporal observation uncertainty correlation in the specification of $\mathbf{C}_c$ (inversion S3), an inversion where no spatial uncertainty correlations were assumed for $\mathbf{C}_{s_0}$ (inversion S4), and an inversion which assumed no uncertainty correlations in the specification of $\mathbf{C}_{s_0}$ and $\mathbf{C}_c$ (inversion S5).

We tested what would happen if observation error correlations were set at seven hours (inversion S6) instead of one hour, as was set for the reference inversion. A one hour observation error correlation lengths results in nonzero off-diagonal covariance terms for up to approximately seven hours from the observation. Assigning a seven hour correlation length resulted in non-zero covariances extending through to at least a day away from the observation.

We also considered inversions where the prior fossil fuel flux uncertainty was doubled (inversion S7) and where it was halved (inversion S8), and similarly for the NEE flux uncertainties (inversions S9 and S10). By doubling or halving the uncertainty of the fossil fuel or NEE component of the total flux, we changed the relative uncertainty contribution each of these made to the total uncertainty when compared with the reference inversion.

Due to the large impact that the estimation of the domestic fossil fuel emissions had on the temporal profile of the total fossil fuel fluxes, we considered a modification of the estimated domestic emissions in the inventory product. In the reference inversion 75% of the domestic emissions from heating were assumed to take place during the six winter months. We tested the impact of this assumption by altering the domestic emissions so that they were distributed uniformly through time, but still spatially distributed according to the population size. This changed the prior estimates of the fossil fuel fluxes and their distribution through time, as well as their uncertainties, which were set at 60% of the domestic emission estimate (inversion S11).

Due to the large uncertainty in the modelling of NEE (Zhang et al., 2013; Moncrieff et al., 2015), particularly over the fynbos biome, we considered that perhaps the average of the NEE estimates from CABLE over the domain may be a more reliable representation of the true flux compared with the pixel-level estimates. Therefore we averaged the NEE and NPP estimates from CABLE over the inversion domain and assigned this average NEE – and NPP for its uncertainty – as the prior biogenic flux estimates (inversion S12).

We considered an inversion where the uncertainties in $\mathbf{C}_c$ were set at $2\,\mathrm{ppm}$ for the day and $4\,\mathrm{ppm}$ at night (inversion S13), excluding the additional components for the error due to wind speed and observation variability that were used in the reference inversion. In this case all the errors in the modelled concentrations are contained within these values, and we disregard the climatic conditions under which the measurements were taken. We tested the impact of increasing the night-time uncertainty in the observation errors to $10\,\mathrm{ppm}$ (inversion S14). We further simplified $\mathbf{C}_c$ by using the simplified uncertainties of $2\,\mathrm{ppm}$ for the day and $4\,\mathrm{ppm}$ at night and also set the temporal observation uncertainty correlation to zero (inversion S15).

### 2.2.4 Alternative control vectors

As a sensitivity analysis we examined two alternative approaches to the control vector. If we assumed that neither the NEE nor fossil fuel flux would change very much from week to week, an option would be to solve for the mean of the six individual fluxes over the four weeks in a given month. We therefore considered a sensitivity test where the inversion solved for one average day and one average night NEE flux within each pixel, and four fossil fuel mean weekly fluxes (day and night working week, day and night weekend) (inversion S16). We also considered performing a separate inversion for each week; i.e. four separate weekly inversions in place of each of the monthly inversions (inversion S17). In this case only the concentration measurements for one week were used and the individual weekly fluxes (two NEE and four fossil fuel) were solved for, and this was repeated for each of the four weeks in the month. The benefit of these two alternative control vectors is that for each individual inversion the resulting $\mathbf{C}_{s_0}$ matrix is much smaller compared with the reference case.

When solving for only one week, or a mean weekly flux for a particular month, the number of surface sources reduced to $10{,}201 \times 6 = 61{,}206$. Solving for individual weeks required $4 \times 2$ additional boundary concentrations to be added to the control vector, and when solving for the mean weekly flux for the month, we allowed the boundary concentrations to differ for each week, and therefore we still solved for the 32 boundary concentrations as in the reference case. Therefore the $\mathbf{C}_{s_0}$ for these two alternative control vectors is 16 times smaller than that of the reference inversion.

The benefit of these two alternative approaches is a substantial reduction (at least 75% reduction) in the time taken to perform the inversion. If the results are similar to that of the reference inversion, this type of saving in the computational time and resources would allow more components of the inversion to be tested in a shorter period of time.

### 2.2.5 Sensitivity analysis approach

A description of the sensitivity tests are presented in Table 1.

The modelled concentrations from each inversion were compared with the observations by assessing the bias and standard deviation of the prior and posterior modelled concentration residuals. Residuals in the prior modelled concentrations were calculated as:

$$\mathbf{c}_{res\,prior} = \mathbf{c} - \mathbf{c}_{mod\,prior}. \tag{14}$$

Residuals in the posterior modelled concentrations were calculated as:

$$\mathbf{c}_{res\,post} = \mathbf{c} - \mathbf{c}_{mod\,post}. \tag{15}$$

where $c_{mod\,prior}$ are the $CO_2$ concentrations modelled from $s_0$ and $c_{mod\,post}$ are the $CO_2$ concentrations modelled from the posterior estimate of $s$, and $c_{res\,prior}$ and $c_{res\,post}$ are the respective residuals in the modelled concentrations. The bias, calculated as the mean of these residuals, and standard deviation of these residuals were provided for each inversion. We plotted the time series of the observed and modelled concentrations to assess the skill of the inversion to reproduce the observed

concentrations, particularly "local events", which were periods of larger than normal spikes in the observed concentration signal. These are presented in the supplementary material (Section 1.3) for all the sensitivity tests.

The posterior fluxes from each inversion were compared with those of the reference inversion in a number of ways. The posterior flux estimates and their spatial distribution were assessed for each inversion by mapping the mean total weekly flux within each pixel for two months (May and September 2012). We calculated the total flux over the domain, and plotted these

weekly total fluxes over time together with the uncertainty bounds. We also considered the total flux over the domain for each month. These total flux estimates are the net flux resulting from the fossil fuel and NEE flux estimates solved for by the inversion. The inversion induces negative correlations between the fossil fuel and NEE flux components from the same week and pixel. When the total flux is considered in a particular pixel, the uncertainty for the total flux will be lower than the sum of the uncertainties for the individual components due to the negative covariance terms. The size of these negative covariances

will depend on the prior information specified in the inversion framework. The total estimate gives an indication of the central tendency, which we can compare between inversions, and allows us to assess, for example, if the inversion is predicting the region to be a net source or a net sink. The uncertainties of these posterior total estimates allow us to assess the confidence we can place around these totals, and how this compares to the estimate itself.

In order to assess the suitability of the prior uncertainty estimates contained in $\mathbf{C}_c$ and $\mathbf{C}_{s_0}$, the $\chi^2$ statistic as described

in Tarantola (2005), was calculated (see equation 13). We compared these statistics between the different inversions to assess the suitability of the uncertainties prescribed to the prior fluxes. Due to the adjustments made, particularly in cases where the uncertainty covariance matrices were simplified, it was expected that some of the inversions would have $\chi^2$ statistics that deviated from one. We chose not to make additional changes to the sensitivity test inversions to improve these statistics, as it would then not be possible to attribute the sensitivity of the inversion solution between the adjustment tested and the additional

adjustment made to the covariance parameters to improve the statistical consistency of the inversion. The number of degrees of freedom of the $\chi^2$ statistic can be divided into the degrees of freedom for signal (DFS) and degrees of freedom for noise (Rodgers, 2000). The DFS describes the number of independent pieces of information provided by the measurements. The DFS were calculated for the first week of March 2012 for the reference and sensitivity test inversions. These statistics are provided in the supplementary material Section 1.1 Figure S1.

**Table 1.** Description of sensitivity tests performed on the Cape Town inversion. Only those aspects which are changed for the sensitivity test are indicated. Other fields are the same as those for the reference inversion.

| Sensitivity test abbreviation | Prior NEE product | Prior Fossil fuel product | NEE error correlations | Observation error correlations | Fossil fuel uncertainties | NEE uncertainties | Observation errors | Control vectors |
|---|---|---|---|---|---|---|---|---|
| S0 | CABLE | Cape Town Inventory | Balgovind 1 km | Balgovind 1 hr | Cape Town Inventory Errors | CABLE NPP | 2 ppm (day); 4 ppm (night) with wind condition and measurement variance inflation | Six individual weekly fluxes |
| S1 | Carbon Assessment Product | | | | | Carbon Assessment NPP | | |
| S2 | | ODIAC | | | ODIAC Estimates ×100 % | | | |
| S3 | | | | No observation error correlation | | | | |
| S4 | | | No NEE error correlation | | | | | |
| S5 | | | No NEE error correlation | No observation error correlation | | | | |
| S6 | | | Balgovind 7 km | | | | | |
| S7 | | | | | Cape Town Inventory Errors ×2 | | | |
| S8 | | | | | Cape Town Inventory Errors ×$\frac{1}{2}$ | | | |
| S9 | | | | | | CABLE NPP ×2 | | |
| S10 | | | | | | CABLE NPP ×$\frac{1}{2}$ | | |
| S11 | | Cape Town Inventory with domestic emissions homogenised over the year | | | Cape Town Inventory Errors domestic emissions homogenised | | | |
| S12 | Averaged CABLE weekly estimates over all pixels | | | | | Averages CABLE NPP weekly estimates over all pixels | | |
| S13 | | | | | | | 2 ppm (day); 4 ppm (night) | |
| S14 | | | | | | | 2 ppm (day); 10 ppm (night) | |
| S15 | | | | No observation error correlation | | | 2 ppm (day); 4 ppm (night) | |
| S16 | | | | | | | | Six average weekly fluxes for each month |
| S17 | | | | | | | | Separate weekly inversions |

# 3 Results

## 3.1 Reference inversion

The results of the reference inversion (S0) are explained in detail in Nickless et al. (2018) and are briefly summarised here. The inversion was able to substantially improve the agreement between the modelled and observed concentrations. The inversion made larger changes to the biogenic fluxes than to the fossil fuel fluxes. Over the Cape peninsula region, where observations made at Robben Island viewed CT central business district (CBD) and harbour emissions as well as biogenic fluxes from the Table Mountain and Cape Point National Park regions, fossil fuel fluxes were adjusted by less than 10%, for example an adjustment from (1.00 to 0.91 $kg\,CO_2\,m^{-2}\,week^{-1}$). An exception is the change to a pixel over a petrol refinery where the inversions made a relatively large change, reducing the total emission in the pixel from 9.43 to 6.62 $kg\,CO_2\,m^{-2}\,week^{-1}$ for May 2012 and from 9.38 to 7.24 for September 2012. Biogenic fluxes were made more negative over the CBD region, with a maximum adjustment from -0.04 to -0.37 $kg\,CO_2\,m^{-2}\,week^{-1}$ in May 2012 and from -0.08 to -0.29 in September 2012, and made more positive over the natural areas, but with much smaller adjustments, a maximum adjustment from -0.04 to 0.04 $kg$ $CO_2\,m^{-2}\,week^{-1}$ in May and from -0.11 to 0.08 in September 2012.

The direction of the adjustments to the prior biogenic fluxes indicated that the CABLE model was overestimating the amount of biogenic carbon uptake over natural areas. Dynamic vegetation models have not been able to simulate fluxes over the fynbos biome well (Moncrieff et al., 2015), and so this result was not surprising. Adjustments to the biogenic fluxes were usually small – ranging between -0.001 and 0.003 $kg\,CO_2\,m^{-2}\,week^{-1}$. The inversion was able to make larger changes to the biogenic fluxes than to the fossil fuel fluxes because the prior biogenic flux uncertainties were made large and because uncertainty correlations were specified between the biogenic fluxes, whereas fossil fuel flux uncertainties were assumed to be independent.

Large uncertainty reductions were made over the natural areas bordering on the CBD, particularly over the Table Mountain National Park, and to natural areas near to the Hangklip measurement site, where the uncertainty was lowered by over 50%. Large uncertainty reductions also occurred over agricultural areas to the north of the CBD region. Uncertainty reductions of up to 60% occurred over a few central CBD pixels, but were generally smaller compared with the uncertainty reductions over natural areas, which reached as high as 92%. When aggregating the fluxes over the domain, uncertainties in the prior aggregated fossil fuel fluxes ranged between 1.3 and 1.5 $kt\,CO_2\,week^{-1}$, whereas the posterior uncertainties ranged between 0.9 and 1.5 $kt$ $CO_2\,week^{-1}$. Uncertainties in the prior aggregated biogenic fluxes ranged between 23.6 and 57.3 $kt\,CO_2\,week^{-1}$ and were reduced to 15.8 and 47.1 $kt\,CO_2\,week^{-1}$ after the inversion. The median percentage uncertainty reduction in the aggregated weekly flux was 28.0 % and ranged between 2.3 and 50.5 %, with the largest reduction occurring in March 2012.

By assigning spatial correlation between biogenic flux uncertainties of neighbouring pixels and assuming independent fossil fuel flux uncertainties, we attempted to provide the inversion with additional information to allow it to better distinguish between these fluxes. The inversion induced negative correlation between fossil fuel and biogenic flux uncertainties in the same pixel. We demonstrated that the posterior uncertainty of any linear combination of terms from the control vector of the fluxes (including the difference between fluxes from the same pixel and the sum of fluxes from the same pixel) will always be unchanged or smaller compared with the prior uncertainty of the same linear combination of elements (Jackson, 1979; Jackson

and Matsu'ura, 1985). This means that although negative correlation between the flux components may be introduced through the inversion, the uncertainty in both the difference between fluxes from the same pixel and the total flux within a pixel will be reduced. When we sum all fluxes within the same pixel, the negative correlations created by the inversion resulted in the posterior uncertainty of the total flux being less than the sum of the posterior uncertainty of the individual fluxes. Therefore

there is an advantage to solving for these fluxes separately.

Clearly the inversion result was strongly dependent on the assumptions regarding the prior fluxes and their uncertainties. The results of the sensitivity tests in subsequent sections explore to what degree these assumptions affected the inversion solution.

## 3.2   Sensitivity tests

To assess the sensitivity of the inversion, we have calculated the aggregated posterior flux across the study period and over the

full spatial domain, together with the posterior uncertainty and uncertainty reduction for each of the sensitivity tests, which are presented in Figure 3. The bar charts, also referred to tornado plots, revealed that changing the prior had the largest impact on the resulting posterior fluxes and their uncertainties. Changing to either the ODIAC fossil fuel product or the carbon assessment biogenic fluxes resulted in prior and posterior flux estimates that were much more positive than those for the reference inversion. The inversion appeared to pull the aggregated fluxes towards an ideal position. The reference posterior fluxes were made more

positive compared to the priors, whereas for the alternative prior products, the inversion drove the posterior fluxes to be less positive. It is hence likely (though not certain) that the true flux is sandwiched between these alternative posterior flux solutions.

The aggregated fluxes were strongly sensitive to the uncertainty spatial correlations specified between the biogenic fluxes. Uncertainty correlations in the biogenic fluxes had a large impact on the spatial distribution of the resulting fluxes, and on the degree to which the inversion was able to make changes across the full domain (Figure 3). Eliminating these uncertainty corre-

lations substantially reduced the inversion's ability to make deviations from the prior fluxes. Therefore, under these sensitivity tests, posterior fluxes were very similar to the prior fluxes, and uncertainty reductions were small.

A short temporal correlation length in the observation uncertainties did not have a large impact on the inversion. Increasing these to seven hours led to greater DFS (see supplementary material Section 1.1 Figure S1), but without having an impact on the flux solution or uncertainty reduction. The statistical consistency also fluctuated much more strongly from month to month

when the temporal observation error correlation was larger compared to a one hour correlation length or assuming independent observation uncertainties. With a correlation length of one hour non-zero off-diagonal elements persisted for approximately seven hours, whereas these off-diagonal elements persisted for much longer when the correlation was set at seven hours. Long correlation lengths are likely not realistic as wind fields observed at the measurement station during the day may be very different to those observed in the evening, reducing the chance of consistent errors in concentration.

The sensitivity test with the smoothed prior biogenic flux over the full domain produced the only posterior flux solution that was corrected to be further from the reference inversion posterior. This inversion did not assume any knowledge about the spatial variability in the surface fluxes, but it appears that providing at least some prior knowledge of where biogenic fluxes are likely to occur – at least separating the ocean and terrestrial fluxes – was important for a sensible posterior flux solution. The domain is not fully or representatively sampled by the observations. By providing a blanket biogenic flux prior across the

domain, areas with large expected biogenic fluxes, which were well sampled by the observation network, had priors that were too carbon neutral, and so biogenic fluxes were made more negative, which was propagated through to neighbouring biogenic fluxes, resulting in a posterior aggregated flux solution that was more negative than the prior. A blanket uncertainty estimate was also used, which meant that the uncertainty associated with the ocean fluxes was much larger compared with the reference

5  inversion, allowing the inversion to make relatively large changes to oceanic pixel fluxes close to the measurement sites.

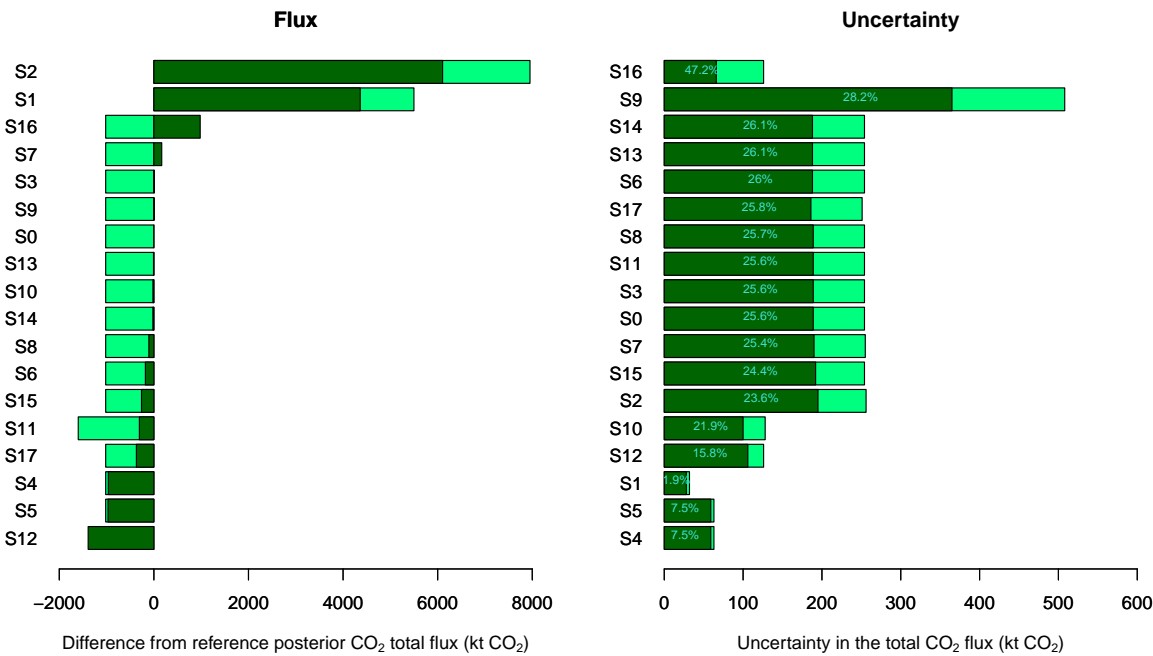

**Figure 3.** Left: Difference between the reference and sensitivity aggregated posterior fluxes over the domain (100 km × 100 km) for the full study period (16 months), ordered from most positive to most negative difference in posterior estimates. The reference inversion posterior aggregated flux was -317 kt $CO_2$. Right: Prior and posterior uncertainties in the aggregated fluxes from reference and sensitivity test inversions. The percentage uncertainty reduction is overlaid over each bar. S0 = Reference Inversion; S1 = Carbon Assessment Inversion; S2 = ODIAC fossil fuel inversion; S3 = Correlation for NEE fluxes only; S4 = Correlation for observation errors only; S5 = No correlation specified in prior covariance matrices; S6 = Long observation error correlation length; S7 = Double fossil fuel uncertainties; S8 = Half fossil fuel uncertainties; S9 = Double NEE uncertainties; S10 = Half NEE uncertainties; S11 = Domestic emission homogenised over the year; S12 = NEE fluxes averaged over the domain; S13 = Simple specification of observation error covariance matrix; S14 = Simple observation error covariance matrix with larger night-time error; S15 = Simple observation error covariance matrix with no correlation; S16 = Inversion solving for mean weekly fluxes over the month; S17 = Separate inversions for each week.

## 3.3 Alternative prior information products

While all the sensitivity test inversions produced prior modelled concentrations that did not track the observations well (see supplementary material Section 1.3 Figures S10 to S27), the carbon assessment and ODIAC prior product inversions (S1 and S2) produced prior modelled concentrations that were on average too large compared with the observed concentrations at both

sites, whereas the reference inversion (S0) underestimated the concentrations at Hangklip and overestimated the concentrations at Robben Island (Figures 4 and 5) (also supplementary material Section 1.5 Figures S37 and S38). The average bias of the prior modelled concentrations from the reference inversion was smaller than the bias for these sensitivity test cases at both sites (see supplementary material Section 1.3 Figures S11 and S12).

The carbon assessment total prior fluxes were notably different to those from ODIAC or the reference inversion. There was

little seasonal variation, with fluxes remaining net positive throughout the study period. The uncertainty bands were very narrow based on the carbon assessment NPP. The mean $\chi^2$ statistic for the S1 inversion of 4.1 (see supplementary material Section 1.2.1 Table S1) indicated that the uncertainties assigned to the fluxes were too small when compared to the uncertainties assigned to the CABLE NEE fluxes in the reference inversion ($\chi^2$ statistic of 1.5 on average), which were closer to being statistically consistent with the assumptions of the inversion. The time series of the prior and posterior fluxes from the S0 and

S2 inversions were more similar to each other over time than to S1, but with the S2 inversion generally having more positive fluxes compared with the reference inversion (Figure 6). These time series indicate that the prior biogenic fluxes drove the temporal variation in the fluxes, whereas the prior fossil fuel fluxes dictated the vertical shift in the flux time series.

The reference inversion generally made fluxes more positive, except for a few winter months when the innovations made fluxes more negative. The S2 inversion had innovations that made the fluxes more negative compared to the priors, except for

September 2012. S1's innovation was to make the fluxes more negative for each month. The magnitude of the innovations were smaller compared to those made to S0 and S2 prior fluxes, limited by the uncertainty placed on the prior biogenic fluxes. For the S1 inversion, both the biogenic flux uncertainties and the correlation lengths were smaller compared to those for S0, and therefore the posterior fluxes were not allowed to differ much from the prior, leaving the modelled concentration residuals before and after the inversion to be very similar, and posterior fluxes almost as uncertain as the prior fluxes.

The spatial pattern in the fluxes (supplementary material Section 1.6 Figures S56 and S57), as reflected in the time series pattern in the weekly fluxes (Figure 6), indicates that prior and posterior fluxes were more positive for the S1 inversion than those of S0 (see also supplementary material Section 1.2.2 Table S2). The spatial heterogeneity in the S1 fluxes was driven by the fossil fuel fluxes, whereas for S0 and S2 this was driven by the biogenic fluxes. The S1 posterior fluxes were largely unchanged from the prior fluxes, except for a notable change made in the September 2012 fluxes where a region of more

negative fluxes was created to the east of the petrol refinery pixel. For the S2 inversion, the ODIAC fossil fuel emissions were highest over the CBD and diminished at distances further from this centre. The spatial distribution of the S0 inversion fossil fuel fluxes were strongly dependent on the transport network and several point sources. The posterior fluxes around the CBD of the S2 inversion were less radial than those in the prior, taking on a spatial pattern more similar to the reference inversion.

With regards to the uncertainty reduction, the S0 inversion was able to obtain higher reductions than either S1 or S2 (Figures 6 and 3, 25.6% reduction compared to 11.0% and 23.6% respectively). The spatial pattern of uncertainty reduction was similar between S0 and S2, whereas S1 showed no uncertainty reduction across much of the domain (see supplementary material Section 1.6 Figures S54 to S59).

Altering the domestic fossil fuel emissions to be the same over time in S11 had little impact on the inversion results when compared with the wholesale change in the prior product. On the other hand, smoothing the biogenic emissions over space in the extreme manner where it was assumed NEE fluxes were the same throughout the domain (S12) had a large impact on the inversion. This resulted in the only inversion where the aggregated fluxes became more negative. The uncertainty reduction was also small (Figure 3). This represents a fairly extreme change to the assumption regarding the spatial distribution of the NEE fluxes, and illustrates the sensitivity of the inversion to the prior information on where fluxes are taking place. In the supplementary material we include timeseries plots of the concentration contributions attributed to the fossil fuel and biogenic fluxes for all the sensitivity test inversions during the month of May 2012 (supplementary material Section 1.4). Robben Island sees far less of the biogenic influence than Hangklip, so in order to make the modelled concentrations more consistent with the observations, the fossil fuel fluxes were adjusted by the inversion, leading to similar contributions to the concentration from biogenic fluxes before and after the inversion. This was the case for the reference inversion S0 and all other inversions except S12, where the inversion made adjustments to the biogenic fluxes instead of the fossil fuel fluxes in order to reduce the modelled concentrations for Robben Island.

Due to the small number of observations relative to the number of sources solved for in the inversion, it is unsurprising that the posterior solution is strongly dependent on the prior information. The results do show that the inversion brings these different prior estimates closer to each other, and therefore the inversion does assist in taking any selected prior closer to the true state, but this is limited by the assumed uncertainty limits placed on the priors, as demonstrated in the S1 inversion.

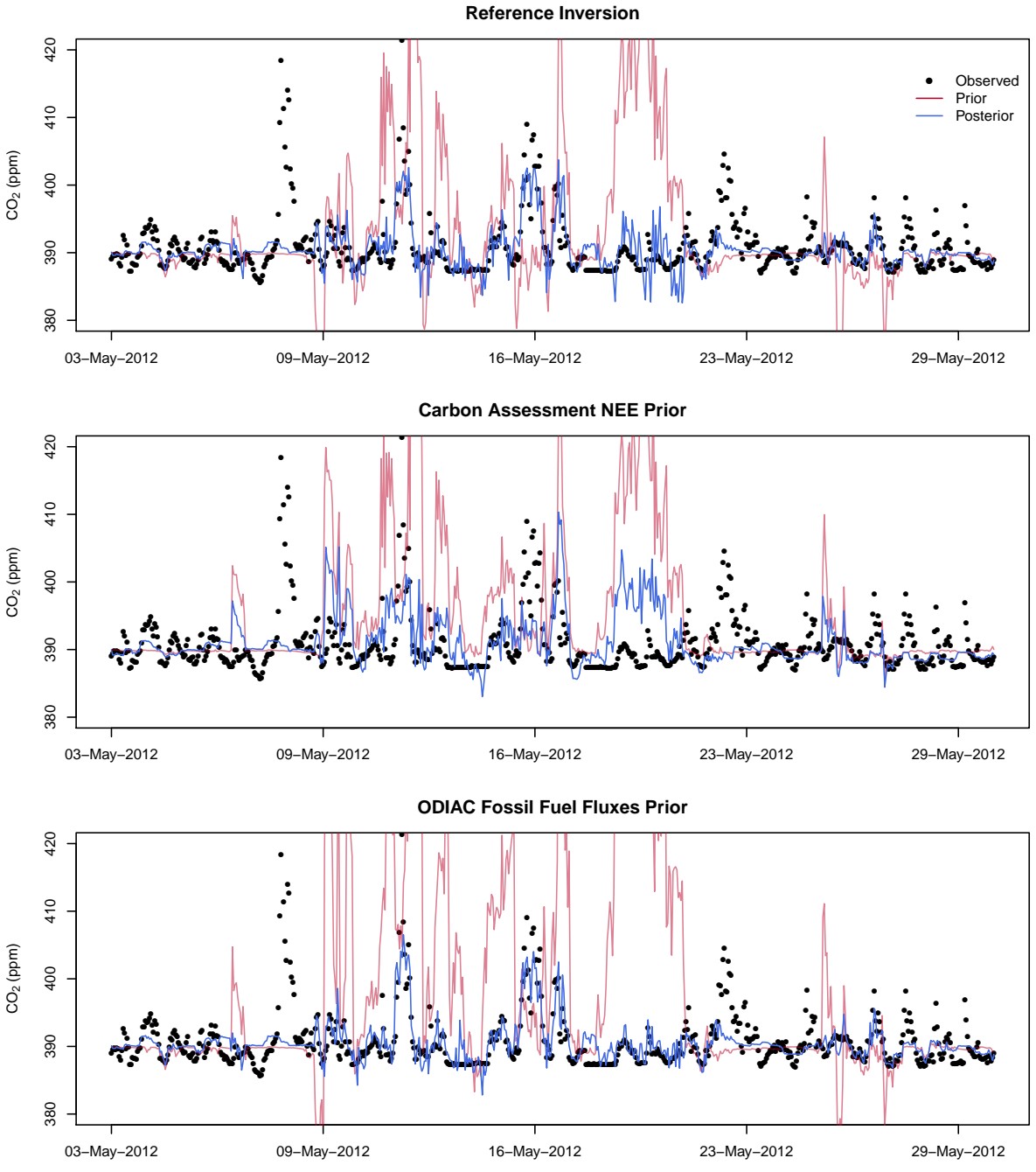

**Figure 4.** Prior and posterior modelled concentrations for the Hangklip site for the month of May 2012 for the reference inversion (top), carbon assessment inversion (middle), and ODIAC fossil fuel flux product inversion (bottom).

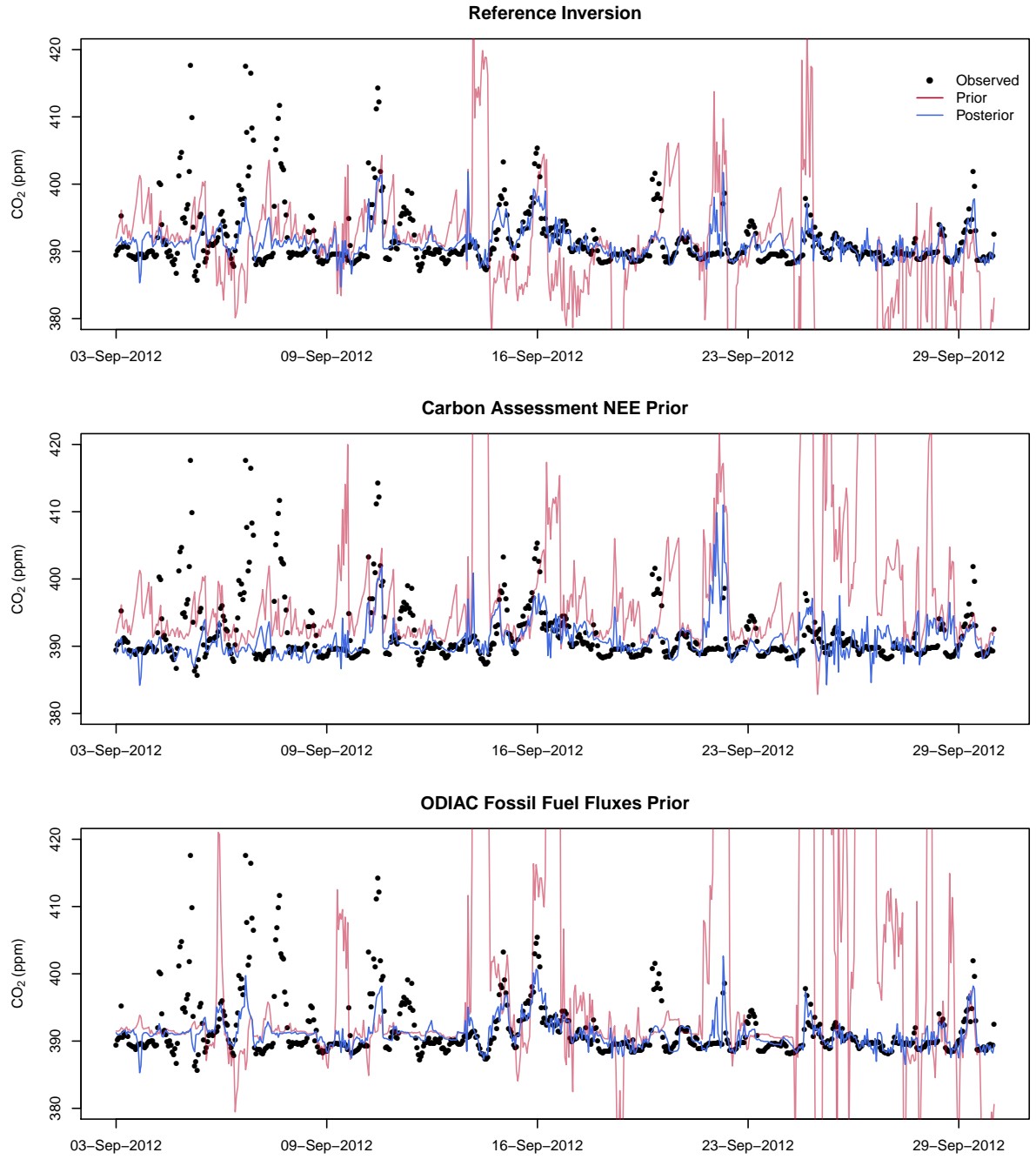

**Figure 5.** Prior and posterior modelled concentrations for the Robben Island site for the month of May 2012 for the reference inversion (top), carbon assessment inversion (middle), and ODIAC fossil fuel flux product inversion (bottom).

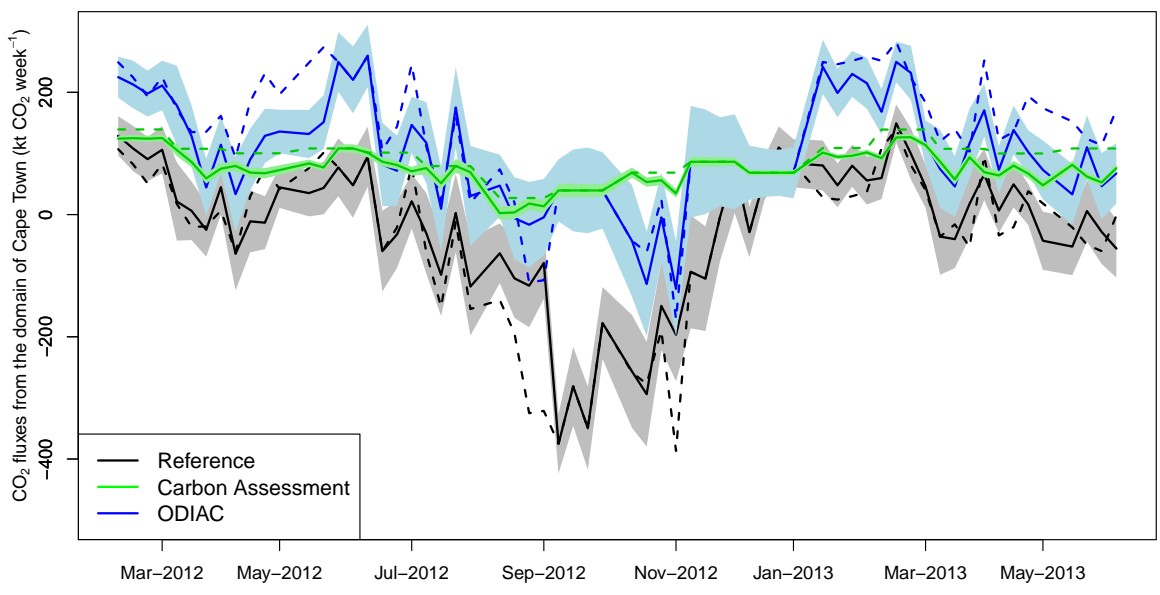

**Figure 6.** Prior and posterior aggregated weekly fluxes over the inversion domain from March 2012 to June 2013 for the reference, carbon assessment and ODIAC inversions. The dashed line represents prior flux estimates and the solid line represents posterior flux estimates.

## 3.4 Uncertainty covariance matrices

The inversion solution was sensitive to the uncertainty spatial correlations assigned to the prior biogenic fluxes. This impacted on the spatial distribution of the fluxes, the magnitude of the total aggregated flux, and the uncertainty reduction achieved by the inversion. By not accounting for the spatial correlations in the biogenic flux uncertainties, this led to uncertainties that were too small, illustrated by average $\chi^2$ statistics above 2 for inversions S4 and S5, which set the spatial correlation of the uncertainties in the biogenic fluxes to zero (see supplementary material Section 1.2.1 Table S1). These inversions also showed little innovation or uncertainty reduction in comparison to the reference, leaving the posterior fluxes to be similar to the priors (Figure 7). This is also reflected in the aggregated fluxes over the study period for S4 and S5, as posterior fluxes were similar to the prior aggregated fluxes and uncertainty reductions in these aggregated fluxes were small. Aggregating over the study period led to posterior flux estimates of -317 and -310 kt $CO_2$ for S0 and S3, whereas S4 and S5 had estimates of -1281 and -1287 respectively, close to the prior estimate of -1336 kt $CO_2$. Uncertainty reductions were reduced from 26.6% to 7.6% when biogenic flux uncertainty correlations were removed.

In comparison, the removal of the temporal correlation in the observation errors in S3 had only a small penalty in the $\chi^2$ statistic. The spatial distribution of the fluxes and uncertainty reductions achieved remained similar to the reference inversion S0 as well. Increasing the temporal correlation length in the observation errors from one hour to seven hours for the S6 inversion had little impact on the posterior flux estimates or the uncertainty reduction achieved, with a posterior aggregated flux over the study period of -497 kt $CO_2$ compared with -317 for S0. The $\chi^2$ statistic was substantially increased to 7.3 on average, and varied more between months compared to all other inversions. Simplifying the observation errors so that they no longer included terms that depended on the meteorological conditions at the site or on how variable the high frequency measurements were during a given hour (S13 to S15) had very little impact on the inversion results.

As the flux uncertainties had already been scaled for the reference inversion to improve the statistical consistency of the uncertainty covariance matrices, it was expected that the $\chi^2$ statistic would be too large for inversions where the uncertainties were halved. This was particularly the case for the biogenic flux uncertainties (S10), as these fluxes were throughout the domain whereas the fossil fuel fluxes were assigned to a smaller part of the domain. Halving or doubling the prior biogenic flux uncertainty (S9 and S10 respectively) led to posterior uncertainties that were roughly half or double the total posterior uncertainty of the S0 inversion, whereas halving or doubling the fossil fuel flux uncertainties (S7 and S8 respectively) made little change to the uncertainty reduction. On the other hand, changing the fossil fuel uncertainties (S7 and S8) had a larger impact on the aggregated posterior flux (-423 kt $CO_2$ when halved and -151 when doubled), compared with changing the biogenic flux uncertainties (S9 and S10), where posterior fluxes remained similar to those obtained by S0. Doubling the fossil fuel flux uncertainty led to generally more positive fluxes across all months.

The spatial distributions of the posterior fluxes in this group of sensitivity tests (S7 to S10) were similar to that of the reference inversion S0. A notable feature in the September 2012 posterior fluxes is that when NEE uncertainties were doubled the inversion was able to reduce the aggregated flux with respect to the priors by creating a region of negative flux in an area close to the oil refinery point source to the north of the CBD region (see supplementary material Section 1.6 Figure S73).

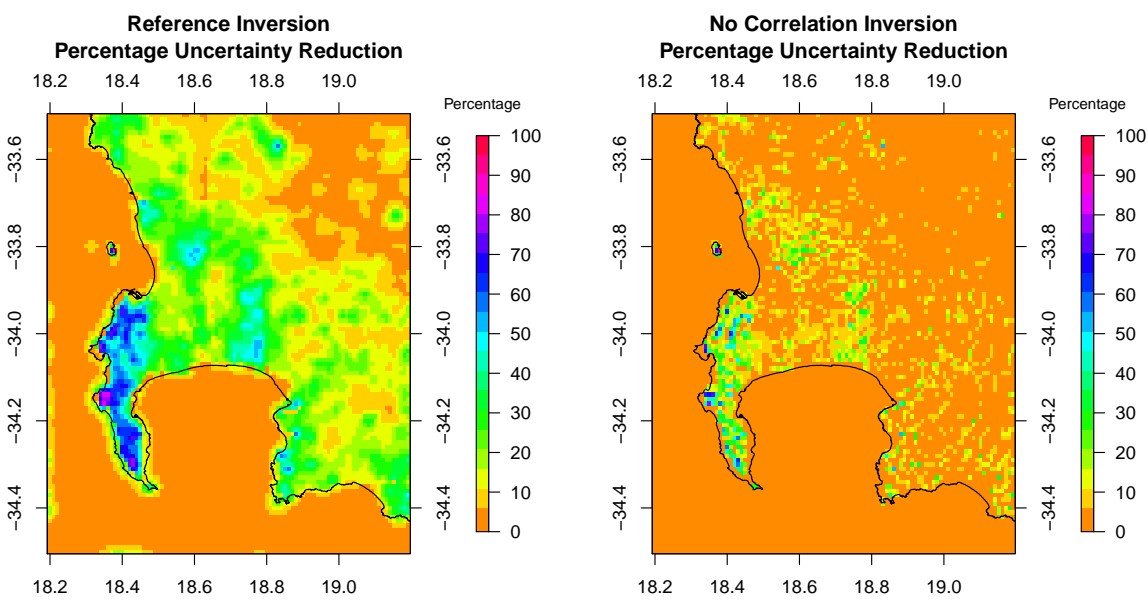

**Figure 7.** Spatial distribution in the pixel-level uncertainty reduction achieved by the inversion to the prior fluxes in May 2012 for the reference inversion (S0) (left), and to the no correlation inversion (S5) (right).

## 3.5 Alternative control vectors

S0 and S17, where separate weekly inversions were performed, had similar aggregated fluxes (Figure 3). For S16, which forced the fossil fuel and biogenic fluxes to be constant over the month, the general pattern over time was similar to S0. For most months the posterior weekly flux was above or below the prior weekly flux to the same degree as S0, but the estimates, as expected, were smoother over time (see supplementary material Section 1.2.3 Figure S9). The monthly aggregated fluxes were generally very close to those from S0 except for August, September and November 2012 (see supplementary material Section 1.2.2 Table S2). These are summer months, and there was a great deal of variation in the aggregated fluxes from week to week in the S0 inversion during these months. S16 generally had aggregated fluxes that were closer to zero than S0 or S17. This had a large impact on the aggregated flux over the full measurement period, due to these less negative posterior aggregated fluxes during the summer months. The aggregated flux for S16 was $662 \, \mathrm{kt} \, CO_2$ compared with the $-317 \, \mathrm{kt} \, CO_2$ for S0. S17 had an aggregated flux of $-687 \, \mathrm{kt} \, CO_2$. This discrepancy is partly due to some weeks with missing observations. In S0 these fluxes would have been adjusted by the available observations for neighbouring weeks, but were completely unconstrained by the observations in S17. The uncertainty reduction in the aggregated estimates was almost double for S16 compared with S0 and S17.

The spatial distribution of the posterior fluxes was very similar for S0 and S17 (see supplementary material Section 1.6 Figure S89), but was distinctly different for S16. Notably, the area around the oil refinery pixel was adjusted to negative fluxes for the month of September (Figure 8). Other areas were made closer to zero compared with S0. The uncertainty reductions at the pixel-level were large for the S16 compared with S0, with more areas of large uncertainty reduction. In particular, the areas of uncertainty reduction above 90% that were restricted to the area over Table Mountain National Park in S0 were now extended over the CBD area.

Consequently the aggregated fluxes for S16 had uncertainty reductions that were twice as large as those for S0, and uncertainties in the aggregated fluxes were much smaller. For the aggregated flux over the full period, the posterior uncertainty was $66 \, \mathrm{kt} \, CO_2$ for S16, compared with the uncertainty of 189 and $186 \, \mathrm{kt} \, CO_2$ from S0 and S17 respectively (Figure 3).

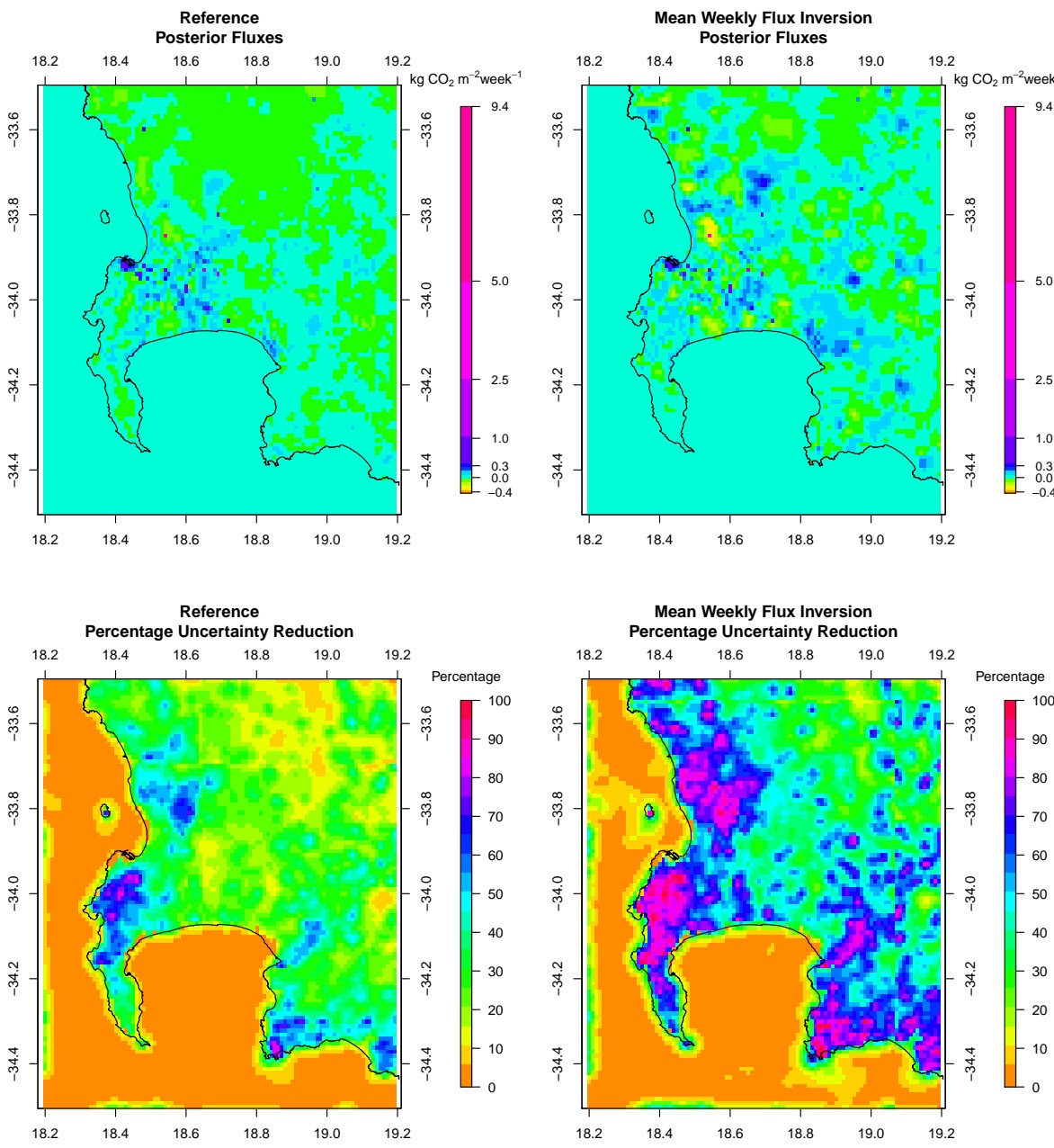

**Figure 8.** Spatial distribution of the posterior fluxes and uncertainty reductions achieved by the reference inversion S0 and mean monthly flux inversion S16 for September 2012.

## 4 Discussion

### 4.1 Alternative prior information products

As Robben Island is dominated by fossil fuel influence from the Cape Town metropolitan area, and Hangklip by biogenic sources from natural and agricultural areas in its vicinity, the discrepancy in the modelled concentrations relative to the observations suggested that the fossil fuel fluxes provided by the prior products are too large in magnitude, and CABLE estimated too much carbon uptake by the biota around the Hangklip site. In the case of the carbon assessment inversion, the bias in the prior modelled concentrations was positive compared with the negative bias of the reference inversion, indicating that the carbon assessment product was underestimating the uptake by the biota. The direction of the correction to the prior fluxes made by the inversion using NEE fluxes from the carbon assessment product suggested that the amount of carbon uptake was insufficient. The NEP fluxes were also smaller compared to those from CABLE, leading to uncertainties that were too small, and therefore an ill-specified inversion. The inversion could not correct the fluxes sufficiently so that modelled concentrations could match better with observed concentrations, and therefore certain localised events (i.e. spikes in the $CO_2$ signal) were not well represented in posterior fluxes from the carbon assessment inversion.

The comparison of inversion results using different prior products provides useful information regarding which direction the true flux estimates are likely to be. A pixel within the CBD limits had similar fossil fuel flux estimates from the ODIAC product compared with the reference inventory product, but the ODIAC product had emissions that were more widespread across the domain away from the CBD. This led to aggregated estimates that were larger under the ODIAC inversion than the reference inversion. Compared to the reference, the ODIAC inversion attempted to reduce the aggregated flux for most months – and to a greater degree – to better match the observations, indicating that compared with the reference inventory, the ODIAC prior was most likely overestimating the amount of fossil fuel emissions from Cape Town to a greater extent for most parts of the study period. When the two prior information products provide divergent prior flux estimates, such that the inversion reduced the flux for one product but increased the flux for the other, it suggests that the true flux lies somewhere between the posterior flux estimates from these two inversions. When the posterior aggregated flux was made smaller than the ODIAC prior but larger than the reference prior aggregated flux, such as during February and March 2013, the true aggregated flux should lie between these two posterior estimates. When the posterior flux was made smaller than the prior for both inversions, we could deduce that the true aggregated flux must be below the minimum of these two posterior estimates, and if we have accurate uncertainty estimates, the true flux should be no smaller than the lower uncertainty limit. Making use of the posterior uncertainties and the direction away from the prior in which the inversions made corrections, a region is suggested where the true flux is most likely to lie (Figure 9). For the CT domain, the inversion results suggest that over the spatial domain investigated, the flux is close to carbon neutral for the majority of the year.

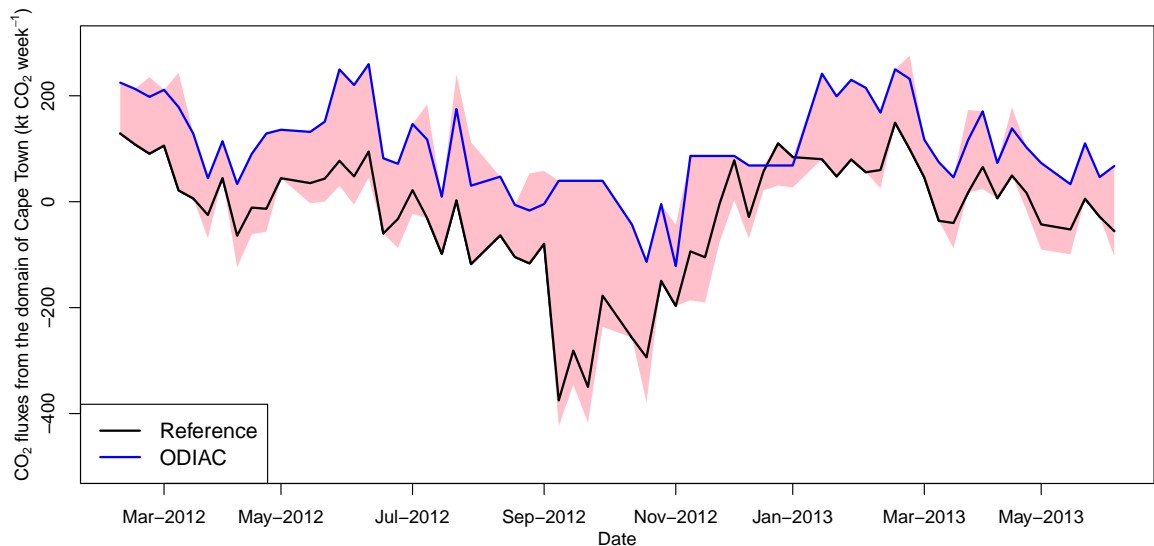

**Figure 9.** Using the posterior estimates of the reference and ODIAC inversions (S0 and S2) and the direction of change from the prior estimate, a region is inferred where in the true aggregated flux is expected to lie, indicated by the pink shaded area.

## 4.2 Uncertainty covariance matrix structure: $\mathbf{C}_{s_0}$ and $\mathbf{C}_c$

From the analysis of the reference inversion (Nickless et al., 2018), the $\chi^2$ statistics indicated that the reference inversion could be improved by small increases to the uncertainty specified in $\mathbf{C}_{s_0}$, either through accounting for a larger correlation length or increasing the pixel-level uncertainties. Removal of the observation error correlations had a very small impact on the goodness-of-fit statistics, or on the posterior flux estimates and uncertainty reduction achieved by the inversion. To ensure that our reference inversion did not deviate too far from conventions for city-scale inversions where observation error correlations are ignored, we assigned a very short error correlation length to the observations of one hour. Although, even with only an hour correlation length, off-diagonal error correlations would have been non-zero for observations at least half a day apart. We considered a longer correlation length in S6, but this had little impact on the inversion and increased the size of the $\chi^2$ statistic, indicating that either the observation errors or flux uncertainties needed to be increased as well to improve statistical consistency. Lauvaux et al. (2009) have shown that observation errors up to 24 hours apart may be strongly correlated. To adequately account for these correlation lengths, a more sophisticated correlation structure may be required where non-zero error correlations are only specified between hours in similar periods of the day, such as afternoon periods for consecutive days, which would be expected to have similar meteorology. The specification of the most suitable observation error length is still under investigation, but the results of these sensitivity tests suggest that this parameter is of less importance than the flux uncertainty correlation lengths.

The impact of the inversion on the posterior fluxes and their uncertainties strongly depended on the specification of the correlation between the uncertainties in the NEE fluxes. In particular, the aggregated fluxes were distinctly different between the reference and test cases ignoring covariances between NEE flux uncertainties, which tended to have aggregated fluxes closer to the priors and uncertainty reductions achieved by the inversion that were much lower (7.6% compared with 26.6% on average by the reference inversion). This indicates that advantage should be taken of knowledge related to the correlation induced by homogeneity of biogenic productivity in subregions of the domain. If this correlation is correctly specified in $\mathbf{C}_{s_0}$, then the inversion is able to make larger adjustments to the prior fluxes and achieve a larger uncertainty reduction in these fluxes.

Specification of the uncertainties in the prior flux estimates is one of the most challenging tasks in an atmospheric inversion exercise. There is little consensus on the correct approach to follow, and it is difficult to ensure that the most important sources of uncertainty are accounted for. The $\chi^2$ statistics indicated that for this Cape Town application, further increasing either the uncertainty in the fossil fuel fluxes or in the NEE fluxes led to statistics closer to one. Increasing the fossil fuel flux or NEE uncertainty led to a lower number of DFS. The degree to which the inversion is constrained by the prior fluxes is inversely related to the specified prior uncertainty. If either the uncertainty in the fossil fuel fluxes or in the NEE fluxes was increased, this led to aggregated flux estimates that were more positive as the inversion was apparently attempting to compensate for the overestimation of the NEE uptake by the CABLE model. When the uncertainties were made smaller, the degree to which the inversion could increase the fluxes was restricted, and the resulting aggregated fluxes were more negative compared with the reference inversion.

An inversion will nudge the flux solution closer to the truth and will always result in reduced uncertainty compared to that which was placed on the prior. If the prior estimates for the fluxes are far from the truth, and the uncertainties are made small, the modelled concentration residuals will be similar before and after the inversion, and uncertainty reduction will be small. Therefore the uncertainties need to be correctly specified to allow the inversion to correct the fluxes as close as possible to the true fluxes. Ideally, large enough to give the inversion the freedom to correct the fluxes towards the truth, but small enough so that the posterior uncertainty is within the required limits. This motivates for the hierarchical Bayesian approach where a distribution is assigned to the uncertainty estimates. It can be shown that in the absence of observation error, doubling or halving the prior uncertainty in the fluxes results in a respective doubling or halving of the posterior uncertainty (see supplementary material Section 1.7). Therefore it us unsurprising that if a prior uncertainty is made larger with respect to a reference inversion specification, that the posterior uncertainty of this inversion will be larger than the posterior uncertainty of the reference.

Normally when an inversion framework is assessed, we are interested in how much uncertainty reduction can be achieved by the available observation network. The uncertainty reduction is dependent on the influence of the observations and on how well the prior information is specified. This set of sensitivity tests demonstrated that if we wish to ensure that the uncertainty bounds around the posterior fluxes are within a prespecified margin, say 10% of the aggregated flux estimate, then we have to ensure that we know enough about the sources such that the prior uncertainty we begin with is sufficiently small. Assuming no large shifts in the mean estimate, it can be shown that if we wish to obtain an uncertainty estimate that is within 10% of the aggregated flux estimate, and we are able to reduce the uncertainty by 25% through the inversion as we have achieved in the Cape Town inversion, then the prior uncertainty estimate would need to be within 13.3% of the prior aggregated flux estimate.

Simplifying the $\mathbf{C}_c$ had very little impact on the inversion results. Increasing the night-time observation errors caused the aggregated flux to be more negative. Assigning an uncertainty in the night-time modelled concentrations of 10 ppm effectively led to the inversion ignoring most of the information available at night, leaving the posterior night-time fluxes (which are mostly affected by the night-time observations) to be similar to their prior estimates. If the inversion is tending to make large corrections to the daytime fluxes, and is now unable to make large corrections to the night-time fluxes, it implies that the aggregated fluxes will be more in error than if the inversion could be constrained by the observations - provided the constraint is good. The analysis of the misfits in the modelled concentrations from the reference inversion (Nickless et al., 2018) demonstrated that the errors in the day and night-time atmospheric transport modelling were not very different, and therefore it is unlikely that assigning errors as large as 10 ppm to all the night-time observations is necessary.

### 4.3 Alternative control vectors

The separate weekly inversions obtained similar results to those of the reference inversion. Therefore, if necessary, for example due to computational costs, the separate weekly inversions could have been performed in place of the monthly inversions used in the reference case. In addition to the reduction in computation resources required, this allows additional features of the inversion to be tested more easily.

The large uncertainty reduction achieved by the solving for a mean weekly flux is expected, as a mean weekly flux estimate over four weeks has four times as many observations to constrain this estimate as separate weekly estimates. The estimates from

the inversion solving for a mean weekly flux were consistent with those from the reference inversion, except in the summer months. During these months observations were often missing. We would expect smaller discrepancies between mean weekly and separate weekly fluxes if data were complete during these periods.

An alternative control vector, which could improve on all three of the alternative control vectors used in this study, would be to solve for separate components of fossil fuel and NEE fluxes. For example, if fossil fuel fluxes were split into those fluxes from sectors which change slowly and those which change more quickly, the inversion could solve for a mean weekly flux over the month for the slow fluxes, and for sectors with faster changes, the inversion could solve for individual weekly fluxes. This would allow greater uncertainty reductions for those fluxes for which a mean weekly flux could be solved, which would in turn reduce the overall uncertainty in the aggregated fossil fuel flux. The NEE flux could also potentially be split into a slow and fast component. The fast component responds to local climate conditions and this component could be tightly constrained by the available climate data. The inversion could solve for the slower component which is much harder to model, allowing this estimate to be constant for a relatively long period, thereby allowing for stronger constraint from the observations.

## 4.4 Inversion sensitivity

If we consider the aggregated posterior fluxes, the variability between flux estimates across those inversions which used the reference control vector is 1962 kt $CO_2$. This is largely driven by the inversions using different prior products, and this variability drops to 469 if these two inversions are removed. It drops further to 375 if the inversions with the transformed prior information are removed. This represents the variability in the aggregated flux estimate across all inversions which used the same prior information products. If we compare this to the uncertainty in the aggregated fluxes, which is approximately 185 kt $CO_2$, it shows that variability between posterior flux estimates from different inversion frameworks is still very large when compared with the uncertainty we expect around the posterior flux estimates. If the inversions with no error correlation between biospheric fluxes are removed, then the variability between inversions drops to 117 kt $CO_2$ – now below the expected uncertainty around the posterior flux from a single inversion. All the inversions that we removed from the estimate of variability were those which had a large influence on the error correlations of the NEE fluxes, either because they were specifically manipulated or because they were affected by the choice of prior product. This demonstrates the important role uncertainty correlations in the prior fluxes have on the posterior flux estimates obtained from an inversion.

Exceptions are the inversions which changed the prior estimates of the fossil fuel fluxes. The fossil fuel fluxes were not assigned uncertainty correlations. Those inversions which altered the prior estimates of the fossil fuel fluxes also had aggregated fluxes that differed when compared with the reference inversion. This is due to the inversion having limited ability to make large changes to the fossil fuel fluxes. The ensemble of posterior fluxes obtained from inversions with alternative prior fluxes allowed us to determine in which direction the inversion was attempting to adjust these fluxes, and provided us with an interval in which we could deduce the true aggregated flux would most likely be located. Changing the control vector also had a large influence on the aggregated flux, but this was largely due to periods with low data completeness.

## 5  Conclusions

Sensitivity tests have shown that to improve the inversion results for the Cape Town inversion, two important advancements should be made to the inversion framework. Firstly the NEE estimates need to be improved. The results from the reference inversion and from these sensitivity tests clearly indicate that CABLE is generally overestimating the amount of $CO_2$ uptake in the domain. Where there is more confidence in the estimation of the biogenic fluxes, either from CABLE for an alternative land-atmosphere exchange model, these reduced uncertainties should be incorporated into the prior information, rather than applying a blanket uncertainty equal to the NPP as done for the reference inversion. For example, over agricultural areas, where the biogenic fluxes may be more reliably modelled, uncertainties may be substantially reduced.

Solving for mean weekly fluxes over a month produced much larger uncertainty reductions. Using an alternative control vector which solves for separate components of the fossil fuel and NEE fluxes that can be split into slow and fast components could take advantage of the larger uncertainty reduction achieved from solving for a mean weekly flux for each month. This could potentially allow the inversion to better distinguish between NEE and fossil fuel fluxes, allowing the inversion to apply corrections to the right flux component (fossil or biogenic), and at the same time obtain aggregated flux estimates with smaller uncertainties than those obtained for the reference inversion. The estimates of the aggregated fluxes were shown to be more reliable in the reference inversion than those for the individual fossil fuel and NEE fluxes (Nickless et al., 2018).

The posterior uncertainties are highly dependent on the prior uncertainties. Of more concern is the large impact that the uncertainty correlation assumed for the NEE fluxes had on the aggregated flux estimates and on the spatial distribution of the posterior fluxes. This has been observed in previous inversions (Lauvaux et al., 2016). Of all the specifications made, the correlation lengths are the most arbitrary, but changing this parameter can entirely alter the distribution of the posterior fluxes. The sensitivity tests suggested that correlations between observation errors were of less importance to the inversion result.

Approaches which allow the data to inform the estimates of the uncertainties and correlation lengths are likely to be more successful at obtaining estimates of the true uncertainty bounds around the inversion posterior flux estimates. Michalak et al. (2005) proposed a maximum likelihood approach to solve for the parameters, and Ganesan et al. (2014) and Wu et al. (2013) proposed an hierarchical Bayesian approach to solve for hyper-parameters of the inversion, including the covariance terms. These approaches have required simplifying assumptions in order to use iterative methods to solve for the parameters, such as assuming the uncertainty is the same across all fluxes or groups of fluxes, or solving for a scaling parameter of the fluxes rather than the fluxes themselves.

These sensitivity analyses performed for this paper did not consider alternative atmospheric transport models. Sensitivity tests on previous city-scale inversions have shown this to be an important source of variation between inversion results (Lauvaux et al., 2016; Staufer et al., 2016; Karion et al., 2019). Future work on the Cape Town inversion will consider alternative regional climate models, such as the WRF (Weather Research and Forecasting model coupled with Chemistry) regional climate model and alternative atmospheric transport models (Karion et al., 2019).

*Data availability.* The hourly $CO_2$ concentration data from Robben Island and Hangklip used for this study are available at https://doi.org/10.17605/OSF.IO/RCFQ4. Data from Cape Point are available at https://www.esrl.noaa.gov/gmd/aero/net/cpt.html.

*Author contributions.* AN installed and maintained all the instrumentation at Robben Island and Hangklip, obtained the measurements and processed these into hourly concentrations, ran and processed the result of the LPDM in Fortran, produced all code and ran the inversion in
Python, processed all the inversion results using R Statistical Software, produced all graphics and tables, designed the sensitivity tests, and was responsible for the development of the manuscript which forms part of her PhD. PJR was the main scientific supervisor, oversaw all implementation of the inversion, and provided guidance on the presentation and interpretation of results. FE performed the coupled CCAM-CABLE simulations. BE provided guidance on statistical issues. RJS provided guidance on the location of the sites and provided input on the interpretation of the results. All authors had the commented on the manuscript.

*Competing interests.* The authors declare that they have no conflict of interest

*Acknowledgements.* We would like to acknowledge and thank Dr Casper Labuschagne, Ernst Brunke and Danie van der Spuy of the South African Weather Service for their assistance in maintaining the instruments at Robben Island and Hangklip, and Dr Casper Labuschagne for his guidance on processing the instantaneous $CO_2$ concentration data; Dr Martin Steinbacher for providing guidance and schematics on the calibration system used on the Picarro instruments; Robin Poggenpoel and Jacobus Smith of Transnet for allowing us access to
the lighthouses; Peter Saaise of Transnet, the Robben Island lighthouse keeper, (and his daughter) for assisting when the instrument was not responding and for tolerating the disturbance to the tranquil lighthouse environment; Dr Marek Uliasz for providing us access to the code for his LPDM model; Dr Thomas Lauvaux for providing guidance on processing the LPDM results and useful discussion on the boundary contribution in the inversion. Use was made of the University of Cape Town ICTS-HPC cluster. Please see http://hpc.uct.ac.za/ for details. We would like to thank Andrew Lewis of the University of Cape Town HPC facility for providing useful advice on improving
the efficiency of the Python runs. This research was funded by competitive parliamentary grant funding from the Council of Scientific and Industrial Research awarded to the Global Change Competency Area towards the development of the Variable-resolution Earth System Model (VRESM; Grants EEGC030 and EECM066). Additional funding was obtained from the South African National Research Foundation for the Picarro instrumentation.

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
