# Peer review of "An atmospheric inversion over the city of Cape Town: sensitivity analyses"

_Atmospheric Chemistry and Physics, 2018_

## Referee Comment (RC1) · Anonymous Referee #1 · 24 Sep 2018

Main comments:

The authors present an atmospheric inversion result over Cape Town focusing on sensitivity analyses related to the technical aspects of the inversion method. I can easily see the authors did a lot of work. However, the presentation needs substantial improvement as well as revisions in technical details.

First, the authors definitely need to rewrite the abstract. Simply it is too long and not organized well (please see my specific comments below).

The introduction section also needs lots of changes or rewriting. Please see my comments below. Basically, it is too technical from the beginning of the section, not providing a gentle overview of the study presented. I recommend that the section be

[Figure]

shortened.

The writing is below average compared to many papers I have reviewed. I understand that the authors did a lot of work but in many places, but the result/discussion presented is not so clear. The paper is too long for the reader to read in current form while there is no exciting scientific findings - this does not mean that the material is not important (it is a different paper). I wonder if the authors can reduce the number of sensitivities cases by (re)moving some of the insignificant results to the supplement.

Please see the detailed comments below and address them before I consider any suggestion for publication.

Detailed comments:

Abstract.

Simply put, the abstract is too long while not conveying useful information in a succinct way. Needs significant improvement in writing (and selecting the most useful pieces of information to be presented here).

Please try to rephrase "A carbon assessment product of natural carbon fluxes, used in place of CABLE, and the Open-source Data Inventory for Anthropogenic CO2 product, in place of the fossil fuel inventory, resulted in prior estimates that were more positive on average than the reference configuration." - a little awkward.

Also, the authors need to divide the following sentences into two (unless made clearer): "For the Cape Town inversion we showed that, where our reference 10 inversion had aggregated prior flux estimates that were made more positive by the inversion, suggesting that the CABLE was overestimating the amount of CO2 uptake by the biota, when the alternative prior information was used, fluxes were made more negative by the inversion. "

Please remove the following (you can state in the results or discussion section): "As the posterior estimates were tending towards the same point, we could deduce that the

best estimate was located somewhere between these two posterior fluxes. We could therefore restrict the best posterior flux estimate to be bounded between the solutions of these separate inversions. "

What is the main conclusion we can gain from the abstract? The authors need to emphasize it. Currently, I only see many small points and cannot determine which one to take home.

P 2, L12: Please remove "where estimates of CO2 fluxes can be derived from measurements of CO2 concentrations at a point location", which does not represent the general atmospheric inversion.

P2, L17: Not all of inversions do that; depends on the study. It could be fossil fuel only.

Introduction: The authors are more focused on the technical aspects of the inversion method considered here by starting describing what atmospheric inversion means in terms of technique, even in the first paragraph of the introduction! Please reframe the introduction so that the authors approach the problem from the urban greenhouse gas (GHG) perspective. People may be interested in Cape Town GHG emissions (more generally), which I haven't heard much before.

Also, please reduce the introduction section because it includes too many technical details/terms. It should be a gentle "introduction" to the paper.

P2, L25: covariance matrices => uncertainty (or error) covariance matrices

P6, L10: "s" should be the surface fluxes, not including the background (i.e., CO2 concentration at the boundary). This is because "Hs" is from the model, not the measurements. Also, c_mod should be Hs_0 (s_0 is prior fluxes in Eq. 1), right?

P6, L13: Change s to s_0. Is s_0 hourly or weekly? Even if you solve for the weekly mean surface fluxes, for CO2, I would expect that hourly prior fluxes were used. Please clarify.

P6, L17: "The boundary concentrations in s"? Why "s" when you talk about concentration. "s" can only be linked to concentrations via H. When you refer to concentrations, it should be "c", not "s"; "s" is fluxes. Right?

P6, L20: Change "can be added to the measurement errors contained in C_c" to "can be added to the error covariance matrix C_c that includes measurement errors". Mathematically, C_c includes all different error sources, but, to be specific/accurate, we want to separate transport errors from those of measurements.

P6, L23: Are 4 and 16 ppmˆ2 the total variance (i.e., including transport error, background error, etc.) in the diagonal elements of C_c that the authors actually used in the inversion? Then, do the authors have any scientific/statistical evidence that these numbers really represent the total irreducible variance in the error covariance matrix? In other words, how did the authors come up with these number?

P7, L3: Why is 1-hour assumed for L? It seems too short. After an hour, are the errors uncorrelated? Usually, following synoptic scales of meteorology, it could go hours and days.

P7, L7: Please add a subsection for the transport model because in current form the authors try to combine the Bayesian inversion method with everything (transport, prior flux, etc.) that is part of the inversion system; not convenient for the reader to follow.

P7, L19: Please add information of temporal and spatial resolutions of the prior flux, as a minimum detail.

P7, L32 - 34: Related to this, please add a few sentences about C_s_0 (prior error covariance) including the structure (e.g., dimension, etc.). In this way, the reader should be able to better understand how the authors treated the prior error covariance.

P8, L2-4: Any concern of aggregation of hourly to weekly? If the authors aggregated into monthly, I would be definitely concerned, but weekly aggregation is in the gray area, it seems to me. The way I would do it is that you still use prior predictions in

hourly (i.e., Hs_0 in hourly in eq. 1) while solving for weekly mean "s". CABLE is originally 1 x 1 km? If not, please say so.

P8, L29: "in place of" => "in addition to". Both bio prior emissions are used?

P9, L6: Where is this standard deviation coming from?

P9, L9: Please add a few sentences about Figure 1. How are the two bio prior fluxes are different (e.g., in total)? How has the uncertainty in the two priors been estimated?

P10, L4-16: This paragraph can be shortened because it does not include any specifics on the author's work. Does Hetia have anything to do with this work? Except for the product description, I don't see any point here.

P10, L21: Please spare your space more on Figure 2 where you compare the two products for prior fossil fluxes. Are they different? If so, how much, in the bottom-up inventory perspective?

P13, L10: The naming is quite confusing. When I started reading the result section, it was confusing and I had to come back here to check the definition. "an inversion which assumed no temporal error correlation in the specification of Cc" := NEE Corr. But no hint of "NEE" in this definition. In Table 1, it says NEE Corr is defined as "no observation error correlation". I understand this is the case without off-diagonal elements. Right? There is a disconnection between L9 and L10 of P13.

P13, L33: What is the difference between "Simp Obs No Corr" and "No Corr". As written, it is not clear.

P14, L1: Please use "state vectors" instead of "control vectors" because "s" (flux) really means the state, which is commonly used in the timeseries model. In GHG inversion work, I have never heard of control vectors.

P14, L11-12: I don't think I have seen a clear description of the background concentration (or boundary concentration). Why only four corners? Since a Lagrangian approach

is used, why not sampling boundary conditions for each of the particles?

Reading "The inversion solved for $4 \times 2 \times 4 = 32$ boundary concentrations" I understand that the authors seem to solve (as in "s") for the a single boundary condition for day or night for each week. 4 corners x 2 (day and night) x 4 weeks? Ideally, each (hourly or sub hourly) CO2 observation has to be associated with the boundary condition. It looks like weekly mean boundary conditions were used, which is not quite okay. Only four corners were used? If so, this is too much simplification. Please clarify how the authors treated the upstream boundary conditions.

Even if the authors used a simple one-valued boundary condition for day and night, I am doubtful about the robustness of the estimation of those 32 values of boundary conditions when solved together with "s". In a sense, Bayesian inversions use regularization methods via prior assumptions, which means a state vector of 244,824 (huge) can still be solved with a small number of observations. But here because the authors are solving for hundreds thousands of parameters, the posterior is highly dependent on the prior. Related to boundary conditions, what this means is that the posterior boundary conditions (if the authors really estimated the posterior boundary conditions while doing inversions, not pre-subtracting; please clarify) is significantly affected by the prior. If so, what prior did the authors use for the boundary condition?

P15, L27: It is okay to use $X^2$ for assessing the goodness-of-fit, but please state the assumption related to this test and whether the data used in the inversion meet the test assumptions. Also, state that what $X^2$ results mean. $X^2$ itself does not guarantee the accuracy of the results.

P18, 3. Results: Please add a subsection here; it looks like an introduction to the Results section but it is a mix of many things. I strongly recommend that the authors remove some to other sections or rewrite it. Basically, what is the main topic for this whole page?

P21, L2: Please define bias (obs - model?) if it has not been done somewhere else.

P21, L11: Then what does it suggest? The model (Gaussian here) and data using ODIAC are more consistent . . . ?

P21, L14: That's because the prior uncertainty was extremely small. Is it a correct prior assumption? It is over-confident!

P22, L2: Which uncertainty? Please be specific.

P22, L7: Typically, biospheric fluxes are much more uncertain. This near-zero uncertainty on the prior suggests to me that the prior assumption is wrong.

P22, L9: Before moving to spatial distribution, do we have any conclusion from this time series comparison? What does all this comparison mean?

P27, L12: How small is the $X^2$ value? Ideally $X^1$ should be close to 1. Is it good or bad? This sounds like ignoring temporal correlation is okay?

P27, L13-15: This needs some clarification. What is the difference between Ref with positive covariance (L13) and just Ref (L15). Which one is compared with which one here. This result suggests "no correlation" has a minimal impact on the posterior?

P27, L17 - L22: The author should be able to explain why there is a such a big difference between weekly and monthly. I don't quite understand why.

P27, L23 - 27: The paragraph starts with Ref and NEE Corr and then mixed up with Obs Corr and No Corr. It is really hard to follow; this happens in many places throughout the paper. Not a smooth reading at all.

P27, L 27: This result seems to be important in terms of error reduction. Please add a couple of sentences for this. From Figure 7, I see the central estimates between No Corr and Ref are similar while the error reductions are different.

Section 3.3 & 3.4: I don't have much comment except for the fact that it is somewhat boring to read - please try to convey in a clearer and succinct way!

P38, L9: Please clarify what "and could not react to local climate conditions" means.

P38, L13 - 15: Not clear what the authors mean by "The ODIAC product extended the fossil fuel fluxes much further a field from the CBD region than the reference inventory. This led to aggregated estimates that were much larger under the ODIAC inversion than the reference inversion." How is the first sentence is related to the second sentence? What do the authors mean by the statement in the first sentence?

P38, L15: "The inversion attempted to reduce the aggregated flux" means when the model tries to match the observations?

P38, L18-20: Please provide estimates (in numbers) for both in the text so that the reader can clearly see the likely true emission estimates. Each inversion should have a uncertainty bound and then I don't understand what it means by "a much narrower uncertainty region than for either inversions."

P38, L26-28: 1 hour is too short. It should be useful to see the results based on 6 hours or 24 hours. I expect the length scale would be hours or even a couple of days.

P39, L17: This is not correct. Prior is just prior. Your sampling from a prior distribution with a fixed mean and a fixed covariance is still a priori info. It does not require the prior sample to be accurate.

P39, L19 - 20: This is because your data points are too small compared to the number of parameters to be solved. In other words, your inversion system is more dependent on the prior rather than observations. In this case, the posterior estimate for the individual pixels won't have much constraint; only the regional total emission may be estimated more or less independently, in the best case. From the Bayesian perspective, the only thing you can do is to report what your assumption was, what model was used and what the result is.

P39, L23 - 27: Not a Bayesian way of thinking, subject to criticism from frequentists.

P42, L14 - 15: Since the authors are using an analytical solutions for a Gaussian

likelihood function, they could use a simple maximum likelihood estimator for the length scale.

P42, L25 - 29: Please correct the sentences. Also, I don't know what the authors are trying to say here, except for the fact that a hierarchical approach may be better.

**paperreview**

―――――――――――――――――――――

---

## Referee Comment (RC2) · N.R.P. Harris (Referee) · 24 Oct 2018

***Nickless et al review***

This manuscript describes a sensitivity study of an inversion of CO2 fluxes in and around Cape Town based on measurements at 3 sites. Cape Town is a city with a strong influence from biogenic fluxes and so provides a good case study for separating the anthropogenic influence from the biogenic influence. The main results from the inversion were published in a previous paper (Nickless et al., 2018). This manuscript concentrates on sensitivity studies on various aspects of the inversion, including the priors used for the biogenic and anthropogenic fluxes and the period over which inversions are averaged. This type of sensitivity analysis is undoubtedly important since cities emit such a large fraction of the global CO2, and there is a need to have robust and well understood inversion methodologies.

The paper is however hard to read. This is partly because it is pretty technical material and partly because so much information is included. This makes it difficult for an interested reader, let alone a casual one, to extract the main points, even after a careful reading. I do not get a feel for the main results from reading the abstract and do not think that the introduction sets the scene for the rest of the paper. I should note that the current discussion and conclusions do a better job of this.

Overall, I think the manuscript could be publishable but only after major revision. I am not making many detailed suggestions as I think a considerable amount of work is needed and the first reviewer has made extensive and well thought out comments. My main comments are as follows:

1. The authors should clarify what the main findings are and then decide what material is needed to back that up in the introduction and in the main body of the manuscript. This should provide a firm basis on which to give a good understanding of the uncertainties and the implications described in the conclusions. That should result in a much tighter and probably shorter manuscript whose contents can be reflected in a clear abstract.

2. In deciding what the main points are, the authors should consider whether ACP or GMD is the more appropriate home for the work. The ACP description includes the statement "*The journal scope is focused on studies with general implications for atmospheric science rather than investigations that are primarily of local or technical interest.*" GMD "*is an international scientific journal dedicated to the publication and public discussion of the description, development, and evaluation of numerical models of the Earth system and its components.*" Models include "*geoscientific model descriptions, from statistical models to box models to GCMs.*"

3. I think that moving to GMD would allow the manuscript to be completely focussed on the technical aspects and might well make it easier to prepare.

4. The supplementary material largely consists of a series of plots which I am not sure are helpful, though I could be persuaded. I would think that some of the current paper could be put into a revised and reduced supplementary material.

5. The present tense should be used for all the new results presented here, and the past tense should be used for previous work and much of the description of the measurements. I am not sure if I am typical, but the mixed use of tense misled me on a few occasions.

6. Some comment should be made about the important differences are present in the emissions products in sections 2.2 and 2.3. As it stands, it is hard to know what to keep in mind for later in the manuscript.

7. It would help to have a short summary of the results from Nickless et al (2018) at the start of Section 3.

8. Can percentages be included in the discussion of the changes vs the reference case?

9. The aspect ratio in Figs 3, 4, and 9 should be increased. They are hard to read at the moment.

---

## Author Response (AR1)

Response to Reviewer Comments on Nickless et al.,

Reviewer 1:

Main comments:

The authors present an atmospheric inversion result over Cape Town focusing on sensitivity analyses related to the technical aspects of the inversion method. I can easily see the authors did a lot of work. However, the presentation needs substantial improvement as well as revisions in technical details.

First, the authors definitely need to rewrite the abstract. Simply it is too long and not organized well (please see my specific comments below).

Response: Following a rewrite of the manuscript, the abstract has been substantially modified.

The introduction section also needs lots of changes or rewriting. Please see my comments below. Basically, it is too technical from the beginning of the section, not providing a gentle overview of the study presented. I recommend that the section be shortened.

Response: The Introduction has been substantially rewritten to give a fuller introduction to city-scale inversions, and details of the sensitivity tests have been kept brief. The purpose of the paper is made clearer.

The writing is below average compared to many papers I have reviewed. I understand that the authors did a lot of work but in many places, but the result/discussion presented is not so clear. The paper is too long for the reader to read in current form while there is no exciting scientific findings - this does not mean that the material is not important (it is a different paper). I wonder if the authors can reduce the number of sensitivities cases by (re)moving some of the insignificant results to the supplement.

Response: The manuscript has been rewritten to improve clarity, and more emphasis is given to those aspects to which the inversion was most sensitive, and sections on tests which had little impact on the result of the inversion have been shortened. We feel that it is important to highlight these aspects of low sensitivity, as this is important information for those who may be concerned about these attributes in similar inversion studies.

Please see the detailed comments below and address them before I consider any suggestion for publication.

Detailed comments:

Abstract.

Simply put, the abstract is too long while not conveying useful information in a succinct way. Needs significant improvement in writing (and selecting the most useful pieces of information to be presented here).

Response: The abstract has been rewritten. It is shorter with better explanation of the purpose and greater emphasis on the main result of the paper.

Please try to rephrase "A carbon assessment product of natural carbon fluxes, used in place of CABLE, and the Open-source Data Inventory for Anthropogenic CO2 product, in place of the fossil fuel inventory, resulted in prior estimates that were more positive on average than the reference configuration." - a little awkward.

Response: This is no longer in the abstract. It now reads: "Alternative prior products were considered in the form of a carbon assessment analysis to provide biogenic fluxes and the ODIAC

(Open-source Data Inventory for Anthropogenic CO2 product) fossil fuel product. These were used in place of the reference inversion's biogenic fluxes from CABLE (Community Atmosphere Biosphere Land Exchange model) and fossil fuel emissions from a bespoke inventory analysis carried out specifically for the Cape Town inversion."

Also, the authors need to divide the following sentences into two (unless made clearer): "For the Cape Town inversion we showed that, where our reference inversion had aggregated prior flux estimates that were made more positive by the inversion, suggesting that the CABLE was overestimating the amount of CO2 uptake by the biota, when the alternative prior information was used, fluxes were made more negative by the inversion."

Response: This has been amended to "Where the reference inversion had aggregated prior flux estimates that were made more positive by the inversion – suggesting that CABLE was overestimating the amount of  $CO_2$  biogenic uptake – the carbon assessment prior fluxes were made more negative by the inversion."

Please remove the following (you can state in the results or discussion section): "As the posterior estimates were tending towards the same point, we could deduce that the best estimate was located somewhere between these two posterior fluxes. We could therefore restrict the best posterior flux estimate to be bounded between the solutions of these separate inversions. " Response: This has been amended to: "As the posterior estimates were tending towards the same point, we could infer that the best estimate was located somewhere between these two posterior fluxes". We have not removed the sentence entirely because this is one of the important points we are trying to make from our conclusions.

What is the main conclusion we can gain from the abstract? The authors need to emphasize it. Currently, I only see many small points and cannot determine which one to take home. Response: The main conclusion from this paper are that spatial and temporal correlations in the flux uncertainties can dictate the solution of an inversion, particular in the typical city-scale inversion framework where high-resolution fluxes are solved for in the inversion. We need to take advantage of these uncertainty correlations in order to propagate the information from the observations further into the domain. To the abstract we have added:

"In summary, estimates of Cape Town fluxes can be improved by using better and multiple prior information sources, particularly on biogenic fluxes. Fossil fuel and biogenic fluxes should be broken down into components, building in knowledge on spatial and temporal consistency in these components into the control vector and uncertainties specified for the sources for the inversion. This would allow the limited observations to provide maximum constraint on the flux estimates."

P 2, L12: Please remove "where estimates of CO2 fluxes can be derived from measurements of CO2 concentrations at a point location", which does not represent the general atmospheric inversion. Response: The introduction has been rewritten. This section has now been changed to "Bayesian inverse modelling provides a top-down technique for verifying emissions and uptake of carbon dioxide (CO2) from both natural and anthropogenic sources. It relies on accurate measurements of CO2 concentrations at suitably located sites which can collect information about these sources at different spatial and temporal scales. The concentration measurements on their own are not sufficient to solve for the emission sources as there are many more sources of CO2 than there are measurements of the concentrations. Therefore well-informed initial estimates of the biogenic and anthropogenic emissions are required, together with uncertainty estimates, which are used to regularise the problem."

P2, L17: Not all of inversions do that; depends on the study. It could be fossil fuel only. Response: This has been reworded as above. An inversion does not necessarily need to solve for both, but both anthropogenic and biogenic fluxes need to be taken into account (either through design, such as limiting he period over which the inversion is performed to be during the dormant season, or by setting one of these components as fixed, or solving for both components of the total flux).

Introduction: The authors are more focused on the technical aspects of the inversion method considered here by starting describing what atmospheric inversion means in terms of technique, even in the first paragraph of the introduction! Please reframe the introduction so that the authors approach the problem from the urban greenhouse gas (GHG) perspective. People may be interested in Cape Town GHG emissions (more generally), which I haven't heard much before. Response: The introduction has been rewritten to give a fuller introduction to the use of inverse modelling for the purpose of solving for greenhouse gas fluxes at the city-scale. The details on the sensitivity analysis in the introduction have been kept light. A summary of the Cape Town reference inversion results have been included at the beginning of the Results section.

Also, please reduce the introduction section because it includes too many technical details/terms. It should be a gentle "introduction" to the paper. Response: The technical details in the introduction have been reduced.

P2, L25: covariance matrices => uncertainty (or error) covariance matrices Response: Corrected

P6, L10: "s" should be the surface fluxes, not including the background (i.e., CO2 concentration at the boundary). This is because "Hs" is from the model, not the measurements. Response: In our inversion framework, the sources include the concentrations at the boundary, which is possible as shown in Ziehn et al 2014 and Nickless et al 2018. This avoids the need to set as fixed what the boundary concentrations are (which are usually modelled with significant errors), which is usually subtracted from the observed concentrations, and these differences used as the observations in the inversion. Because we worked with a rectangular domain, it made sense to use the boundaries at the four cardinal directions. We do not have modelled concentrations at the boundaries. Instead, we were fortunate to have a GAW measurement site in the domain which observed background conditions for the majority of the time, and due to the homogeneity of the region around Cape Town, these measurements could be taken as representative of boundary conditions on all sides. By solving for the concentrations, but imposing small uncertainties on these concentrations, the inversion can make small corrections to the boundary concentrations, but these corrections would not dominate the inversion solution.

**Also, c\_mod should be Hs\_0 (s\_0 is prior fluxes in Eq. 1), right?**

Response: In this case c\_mod should be equal to Hs. Even if we know exactly what s are, Hs only gives us modelled concentrations, and difference between these modelled concentrations and the true concentrations are then the observation errors.

P6, L13: Change s to s\_0. Is s\_0 hourly or weekly? Even if you solve for the weekly mean surface fluxes, for CO2, I would expect that hourly prior fluxes were used. Please clarify. Response: s has been changed to s\_0. We have provided more details on the Bayesian inversion framework. The fluxes are weekly fluxes. It is possible to calculate a sensitivity matrix to solve for a flux in any time step. We chose to solve for weekly fluxes (i.e. we assumed that the day and night fluxes remained constant over a period of a week) since daily fluxes would lead to much larger matrices than would be manageable in the current framework, and there are not enough observations available to resolve fluxes at an hourly time step.

P6, L17: "The boundary concentrations in s"? Why "s" when you talk about concentration. "s" can only be linked to concentrations via H. When you refer to concentrations, it should be "c", not "s"; "s" is fluxes. Right?

Response: The boundary concentrations are included in s\_0 since we solve for these concentrations in the inversion. Ziehn et al (2014) shows the derivation of the sensitivity matrix for boundary concentrations solved for in a limited domain regional inversion.

P6, L20: Change "can be added to the measurement errors contained in C\_c" to "can be added to the error covariance matrix C\_c that includes measurement errors". Mathematically, C\_c includes all different error sources, but, to be specific/accurate, we want to separate transport errors from those of measurements.

Response: This has been reworded and the section expanded.

P6, L23: Are 4 and 16 ppm2 the total variance (i.e., including transport error, background error, etc.) in the diagonal elements of C\_c that the authors actually used in the inversion? Then, do the authors have any scientific/statistical evidence that these numbers really represent the total irreducible variance in the error covariance matrix?

In other words, how did the authors come up with these number?

Response: These are only the minimum observation errors. Further terms are added for the observation errors based on the observed variability in the measurements at the site within each hour and the average wind speed at the site during each hour. More information has been added to the methods section on the derivation of the observation errors.

P7, L3: Why is 1-hour assumed for L? It seems too short. After an hour, are the errors uncorrelated? Usually, following synoptic scales of meteorology, it could go hours and days.

Response: A 1 hour correlation length leads to non-zero correlations between observations at least seven hours apart. Most city-scale and mesoscale inversions do not include observation error correlations, and work with diagonal matrices, although it is known that observation errors are correlated. We have included an additional case using a 7 hour correlation length, which leads to non-zero correlations between observations further than 24 hours apart.

P7, L7: Please add a subsection for the transport model because in current form the authors try to combine the Bayesian inversion method with everything (transport, prior flux, etc.) that is part of the inversion system; not convenient for the reader to follow.

Response: The description of the inversion framework has been expanded and divided into subsections.

P7, L19: Please add information of temporal and spatial resolutions of the prior flux, as a minimum detail.

Response: These additional details have been added.

P7, L32 - 34: Related to this, please add a few sentences about C\_s\_0 (prior error covariance) including the structure (e.g., dimension, etc.). In this way, the reader should be able to better understand how the authors treated the prior error covariance. Response: These additional details have been added.

P8, L2-4: Any concern of aggregation of hourly to weekly? If the authors aggregated into monthly, I would be definitely concerned, but weekly aggregation is in the gray area, it seems to me. The way I

would do it is that you still use prior predictions in hourly (i.e., Hs\_0 in hourly in eq. 1) while solving for weekly mean "s". CABLE is originally 1 x 1 km? If not, please say so.

Response: Hs\_0, which are the modelled concentrations, is hourly. The sensitivity matrix H relates weekly fluxes to hourly concentrations (i.e. assume that the day and night fluxes remain constant during the week). CABLE was dynamically coupled to the regional climate model, and therefore was driven by inputs on a 1 x 1 km spatial grid. Additional details on the inversion framework have been added.

**P8, L29: "in place of" => "in addition to". Both bio prior emissions are used?**

Response: The sentence referred to here is "We used these estimates of NEE and NPP in place of those from CABLE (inversion Carbon Assess)." The reference inversion used the net ecosystem exchange from CABLE as the biogenic flux prior and the net primary productivity as the estimate of the uncertainty in the biogenic flux. As a sensitivity test, the estimates from CABLE were replaced with those from a carbon assessment product (NEE for the biogenic flux prior and NEP as the uncertainty in this flux).

This sentence has been changed to "As a sensitivity test, the NEE and NPP from CABLE estimates used for the biogenic flux priors and their uncertainties were replaced with NEE and NPP from the carbon assessment product and the inversion rerun with these priors".

**P9, L6: Where is this standard deviation coming from?**

Response: This estimate was calculated as the standard deviation between the fynbos biome pixels from the carbon assessment product. This information has been added. This sentence has been changed to: "The carbon assessment estimated the GPP flux for the year in the fynbos biome to be  $521 \text{ g CO}_2 \text{ m}^{-2} \text{ year}^{-1}$  with a standard deviation of  $492 \text{ g CO}_2 \text{ m}^{-2} \text{ year}^{-1}$  across pixels with 1 km2 resolution."

P9, L9: Please add a few sentences about Figure 1. How are the two bio prior fluxes are different (e.g., in total)? How has the uncertainty in the two priors been estimated? Response: These details have been added in the section on the priors. The uncertainties are taken as the NPP estimate from the products, as has been done in previous mesoscale inversions. We favour this approach over assigning a percentage uncertainty, as biogenic fluxes in many of South Africa's biomes are often close to carbon neutral, resulting from large productivity and respiration fluxes during the growing seasons. Therefore if a percentage uncertainty was assigned to the NEE flux, these uncertainties would be close to zero, which would be unrealistic.

After Figure 1 we have added: "The biogenic  $CO_2$  fluxes are more homogeneous across the domain in the carbon assessment product. This can be explained by the products used as inputs for the estimation of the carbon stock components, such as FAPAR, which would not be expected to differ considerably from pixel to pixel in this domain. CABLE predicts greater  $CO_2$  uptake. The average  $CO_2$  flux over the course of the study period and across the domain, was -41 g  $CO_2$  m-2 week-1 according to the carbon assessment and -172 g  $CO_2$  m-2 week-1 according to CABLE. The true flux is likely to be highly variable but close to carbon neutral over a long period of time (several years)."

P10, L4-16: This paragraph can be shortened because it does not include any specifics on the author's work. Does Hetia have anything to do with this work? Except for the product description, I don't see any point here.

Response: This has been shortened. The discussion on the Hestia product was to show how ODIAC compared to alternative inventory data available in other settings.

P10, L21: Please spare your space more on Figure 2 where you compare the two products for prior fossil fluxes. Are they different? If so, how much, in the bottom-up inventory perspective? Response: This figure has been modified and additional statistics comparing the products have been provided. A paragraph below the figure has been added:

"The ODIAC product gave similar fossil fuel fluxes over pixels in the CBD area compared with the inventory estimates. The inventory estimates were concentrated over the road network, point sources, and areas of high population density, whereas the ODIAC product dispersed emissions over the domain, with an area of high concentration over the CT metropolitan area and decreasing emissions away from this region. The average fossil fuel flux for the domain over the study period was 134 g  $CO_2 m^{-2}$  week-1 according to the inventory and 274 g  $CO_2 m^{-2}$  week-1 according to the ODIAC product."

P13, L10: The naming is quite confusing. When I started reading the result section, it was confusing and I had to come back here to check the definition. "an inversion which assumed no temporal error correlation in the specification of Cc" := NEE Corr. But no hint of "NEE" in this definition. In Table 1, it says NEE Corr is defined as "no observation error correlation". I understand this is the case without off-diagonal elements. Right?

Response: We have decided to use sensitivity case numbers instead, to avoid any confusion. A table is provided that gives the details of each case. Yes, No Corr means that the uncertainty covariance matrix of the fluxes is diagonal.

There is a disconnection between L9 and L10 of P13.

Response: This has been reworded. The sentences have been rewritten as:

"To assess the sensitivity of the posterior flux estimates, their uncertainties, and their distribution in space to the specification of the uncertainty correlations, we ran inversions where the non-zero offdiagonal elements of **C\_s0** and **C\_c** in the reference inversion were systematically set to zero. We considered an inversion which assumed no temporal observation uncertainty correlation in the specification of **C\_c** (inversion S3), an inversion where no spatial uncertainty correlations were assumed for **C\_s0** (inversion S4) and an inversion which assumed no uncertainty correlations in the specification of **C\_s0** and **C\_c** (inversion S5)."

P13, L33: What is the difference between "Simp Obs No Corr" and "No Corr". As written, it is not clear.

Response: Simp Obs is the scenario where the observation error is set as either 4ppm2 or 16ppm2 (excluding the additional components for within-hour measurement variability and within-hour wind speed that were specified in the reference inversion). Both of these cases used diagonal observation error covariance matrices.

To improve clarity, this paragraph has been modified to:

"We considered an inversion where the uncertainties in **C\_c** were set at 2 ppm for the day and 4 ppm at night (inversion S13), excluding the additional components for the error due to wind speed and observation variability that were used in the reference inversion. In this case all the errors in the modelled concentrations are contained within these values, and we disregard the climatic conditions under which the measurements were taken. We tested the impact of increasing the night-time uncertainty in the observation errors to 10 ppm (inversion S14). We further simplified **C\_c** by using the simplified uncertainties of 2 ppm for the day and 4 ppm at night and also set the temporal observation uncertainty correlation to zero (inversion S15)."

P14, L1: Please use "state vectors" instead of "control vectors" because "s" (flux) really means the state, which is commonly used in the timeseries model. In GHG inversion work, I have never heard of control vectors.

Response: The use of control vector is quite commonly used in mesoscale and city-scale inversion studies (e.g. Lauvaux et al 2012 and Oda et al 2017). We would prefer to continue to use control vector to be consistent with the companion paper already published.

We have included a sentence earlier in the manuscript "Additionally we were interested in the composition of the control vector, also referred to as the state vector, which specifies the surface fluxes and domain boundary concentrations to be solved for by the inversion."

P14, L11-12: I don't think I have seen a clear description of the background concentration (or boundary concentration). Why only four corners? Since a Lagrangian approach is used, why not sampling boundary conditions for each of the particles? Reading "The inversion solved for  $4_2_4 = 32$  boundary concentrations" I understand that the authors seem to solve (as in "s") for the a single boundary condition for day or night for each week. 4 corners x 2 (day and night) x 4 weeks? Ideally, each (hourly or sub hourly) CO2 observation has to be associated with the boundary condition. It looks like weekly mean boundary conditions were used, which is not quite okay. Only four corners were used? If so, this is too much simplification. Please clarify how the authors treated the upstream boundary conditions.

Response: We did not have modelled concentrations of  $CO_2$  at the boundaries of the domain. We used the cardinal directions because our limited domain was gridded. We would not expect great variations in the  $CO_2$  concentrations at the boundaries of this domain as there are no close sources either near the ocean borders or the terrestrial borders. The differences between the concentrations at the boundary and the concentration measured at the background site (Cape Point) located within the domain are expected to be very small, certainly smaller than errors in modelling  $CO_2$  concentrations if a chemical transport model had been used.

We have added a full description of the treatment of the boundary concentrations in a new section on the reference inversion. With regards to the sensitivity tests, which is the focus of this paper, all of the inversions used the same prior boundary concentrations and solved for the same 32 boundary concentrations. As these sensitivity tests were focused on the uncertainty covariance matrices, we did not consider any sensitivity tests listed here which changed the way we treated the boundary concentrations, but kept this as a constant between all inversions tested.

Even if the authors used a simple one-valued boundary condition for day and night, I am doubtful about the robustness of the estimation of those 32 values of boundary conditions when solved together with "s". In a sense, Bayesian inversions use regularization methods via prior assumptions, which means a state vector of 244,824 (huge) can still be solved with a small number of observations. But here because the authors are solving for hundreds thousands of parameters, the posterior is highly dependent on the prior. Related to boundary conditions, what this means is that the posterior boundary conditions (if the authors really estimated the posterior boundary conditions while doing inversions, not pre-subtracting; please clarify) is significantly affected by the prior. If so, what prior did the authors use for the boundary condition?

Response: The prior for the boundary condition was the average concentration taken from the background signal at Cape Point during the course of a week. Variations in this concentration are expected to be small during the course of a week, and there are no large nearby sources outside the domain. The concentration at the boundary is solved for in the inversion, but only a small uncertainty is placed on these concentrations, informed by the observed hourly concentrations, which means that the inversion has to correct the modelled concentration predominantly through making changes to the fluxes within the domain. This was shown to be the case in the reference

inversion (Nickless et al. 2018), and a full discussion on the use of this approach is provided in the companion paper. Where the inversion did make corrections to the boundary concentrations, these corrections were usually made to the terrestrial boundary, which is what we expected.

For the purposes of this paper, which focuses on sensitivity analyses, the boundary condition was set to be the same for all cases, therefore for each sensitivity test any sensitivity shown in the inversion solution in comparison with the reference inversion should be due to the adjustment made to the inversion for this test, and not due to the approach used for accounting for the boundary concentrations.

P15, L27: It is okay to use X2 for assessing the goodness-of-fit, but please state the assumption related to this test and whether the data used in the inversion meet the test assumptions. Also, state that what X2 results mean. X2 itself does not guarantee the accuracy of the results. Response: This has been changed to: "In order to assess the suitability of the prior uncertainty estimates contained in **C\_s0** and **C\_c**, the  $\chi^2$  statistic as described in Tarantola (2005), was calculated". More explanation on the statistical assumptions and caveats of this statistic for making this assessment are provided in a new section relating to the use of the  $\chi^2$  statistic for the reference inversion.

P18, 3. Results: Please add a subsection here; it looks like an introduction to the Results section but it is a mix of many things. I strongly recommend that the authors remove some to other sections or rewrite it. Basically, what is the main topic for this whole page? Response: The results section has been rewritten to more succinct and to focus on the main finding of the sensitivity analysis. The description of non-significant tests has been made much shorter. The first section of the results gives a summary of the reference inversion for Cape Town.

P21, L2: Please define bias (obs - model?) if it has not been done somewhere else. Response: The definition for bias has been added in Section 2.2.5.

P21, L11: Then what does it suggest? The model (Gaussian here) and data using ODIAC are more consistent : : :?

Response: This suggests that the uncertainty estimates for the prior fluxes taken from the ODIAC product, which were set at 100% of the ODIAC estimate, are consistent with the statistical assumptions of the inversion. The uncertainties used for the ODIAC product are much larger than those used for the estimates derived from the inventory in the reference inversion. As  $\chi^2$  is not less than one, it indicates that these larger uncertainties are needed in order to adjust the prior flux estimates so that the modelled concentrations better match the observed concentrations.

P21, L14: That's because the prior uncertainty was extremely small. Is it a correct prior assumption? It is over-confident!

Response: The same approach for assigning uncertainties to the prior biogenic fluxes in the reference inversion (using the NPP fluxes as the uncertainty) was applied to the carbon assessment inversion. In this case, the uncertainty estimates are too narrow (if we assume the observation errors are large enough). We wanted to show what the inversion would look like if we swapped out the reference biogenic component for an alternative without making any further changes.

P22, L2: Which uncertainty? Please be specific.

Response: This was referring to the total flux uncertainty. This has been made more specific.

P22, L7: Typically, biospheric fluxes are much more uncertain. This near-zero uncertainty on the prior suggests to me that the prior assumption is wrong.

Response: The uncertainty is not near-zero, but closer to zero than those for the reference inversion. This has been made clearer. It is the difference in the uncertainty from the prior to posterior uncertainty that is small. As you have stated, the prior uncertainties are too small, and therefore the Bayesian inversion has not been able to provide sufficient correction to the prior fluxes, and as a consequence, the difference between the prior and posterior fluxes and the difference between the observed and modelled concentrations are too large and are not centred around zero. Therefore, the  $\chi^2$  statistic is greater than one. The uncertainty in the fluxes after the inversion is almost as great as the uncertainty before the inversion.

P22, L9: Before moving to spatial distribution, do we have any conclusion from this time series comparison? What does all this comparison mean?

Response: The figure for the time series has been changed to one which shows the time series of the posterior flux estimates on one step of axes for all three inversions. This shows better how much each set of posterior fluxes has been adjusted from the prior estimates, and in which direction the inversion has shifted the fluxes. The time series shows that under the carbon assessment inversion, the uncertainty limits are too narrow, and so very little adjustment by the inversion was possible. The width of the uncertainty bounds of the ODIAC inversion was similar to those for the reference inversion. The inversion has shifted the more positive prior fluxes under the ODIAC inversion to be closer to zero, and in the reference inversion, the more negative fluxes have been shifted towards zero as well. The figure of the time series plots suggests that the inversion process is consistently shifting the time series of the prior fluxes towards the same ideal time series.

P27, L12: How small is the X2 value? Ideally X1 should be close to 1. Is it good or bad? This sounds like ignoring temporal correlation is okay?

Response: The temporal observation error correlations did not change the  $\chi^2$  very much, with statistics remaining close to one. Therefore, if it is assumed that the other components of the covariance matrices are correct, then removing these temporal correlations is consistent with the statistical assumptions of the inversion.

P27, L13-15: This needs some clarification. What is the difference between Ref with positive covariance (L13) and just Ref (L15). Which one is compared with which one here. This result suggests "no correlation" has a minimal impact on the posterior?

Response: There is no difference, as Ref contains these positive covariances. It is the test cases Obs Corr and No Corr where these positive covariances were made zero. The sentence referred to here has been corrected. What we meant to say here was:

In the reference inversion the positive covariances specified between neighbouring NEE flux uncertainties led to large prior and posterior uncertainty around the aggregated weekly fluxes. If these positive covariances are removed from **C\_s0** then the uncertainty around the aggregated total flux was much smaller. On the other hand, the test case which retained the positive covariances in **C\_s0** (S3) had uncertainty bounds around the prior and posterior aggregated fluxes that were indistinguishable from those in the reference inversion.

This section has now been shortened to: "In comparison, the removal of the temporal correlation in the observation errors in S3 had only a small penalty in the  $\chi^2$  statistic. The spatial distribution of the fluxes and uncertainty reductions achieved remained similar to the reference inversion S0 as well. Increasing the temporal correlation length in the observation errors from one hour to seven hours for the S6 inversion had little impact on the posterior flux estimates or the uncertainty reduction achieved,..."

P27, L17 - L22: The author should be able to explain why there is a such a big difference between weekly and monthly. I don't quite understand why.

Response: There is no difference between weekly and monthly uncertainty reduction. In this paragraph we have focused on the uncertainty reduction, and this is summarised for each month in the supplementary material (Table S2), and summarised over the whole study period in Table S1. The flux estimates are aggregated over a month (aggregated over space and time). If we look at the relative difference between inversions in the spatially aggregated estimates over a week, this relationship is similar to what we get if we aggregated over the month.

P27, L23 - 27: The paragraph starts with Ref and NEE Corr and then mixed up with Obs Corr and No Corr. It is really hard to follow; this happens in many places throughout the paper. Not a smooth reading at all.

Response: The labelling of the inversions has been changed. These cases are all being referred to here, in this final paragraph of the section, because we intended to compare the inversions which had modified uncertainty correlations, which in the previous version were inversions NEE Corr, Obs Corr and No Corr (now inversions S3, S4 and S5).

P27, L 27: This result seems to be important in terms of error reduction. Please add a couple of sentences for this. From Figure 7, I see the central estimates between No Corr and Ref are similar while the error reductions are different.

Response: We have changed the text here to: "The inversion solution was sensitive to the uncertainty spatial correlations assigned to the prior biogenic fluxes. This impacted on the spatial distribution of the fluxes, the magnitude of the total aggregated flux, and the uncertainty reduction achieved by the inversion. By not accounting for the spatial correlations in the biogenic flux uncertainties, this led to uncertainties that were too small, illustrated by average  $\chi^2$  statistics above 2 for inversions S4 and S5, which set the spatial correlation of the uncertainties in the biogenic fluxes to zero (see supplementary material Table S1). These inversions also showed little innovation or uncertainty reduction in comparison to the reference, leaving the posterior fluxes to be similar to the priors (Figure 7)."

Section 3.3 & 3.4: I don't have much comment except for the fact that it is somewhat boring to read - please try to convey in a clearer and succinct way!

Response: The results section has been rewritten.

P38, L9: Please clarify what "and could not react to local climate conditions" means. Response: This sentence has been reworded. "The direction of the correction to the prior fluxes made by the inversion using NEE fluxes from the carbon assessment product suggested that the amount of carbon uptake was insufficient. The NEP fluxes were also smaller compared to those from CABLE, leading to uncertainties that were too small, and therefore an ill-specified inversion. The inversion could not correct the fluxes sufficiently so that modelled concentrations could match better with observed concentrations, and therefore certain localised events (i.e. spikes in the CO2 signal) were not well represented in posterior fluxes from the carbon assessment inversion."

P38, L13 - 15: Not clear what the authors mean by "The ODIAC product extended the fossil fuel fluxes much further a field from the CBD region than the reference inventory. This led to aggregated estimates that were much larger under the ODIAC inversion than the reference inversion." How is the first sentence is related to the second sentence?

What do the authors mean by the statement in the first sentence?

Response: The ODIAC product has fossil fuel emissions that non-zero for pixels further away from the Cape Town central business district compared with the Cape Town inventory, where the

emissions were localised and concentrated around road networks and point sources, and within regions where the census information located the majority of the population. There are many terrestrial pixels on the outskirts of the domain, near the terrestrial boundaries of the domain where the population size is small, there are no point sources nor a substantial road network, and so the fossil fuel emissions are close to zero. The ODIAC product smoothed the emissions further from the central area, with most pixels having non-zero fossil fuel emissions. If the emissions are aggregated over the domain, the ODIAC product had a larger aggregated flux compared with the Cape Town inventory, and this persisted in the posterior fluxes as well. This is expected as the Cape Town inventory only account for the major point sources in the domain. The aggregation of the smaller point sources that are unknown is almost certainly significant.

We have changed this to: "The comparison of inversion results using different prior products provides useful information regarding which direction the true flux estimates are likely to be. A pixel within the CBD limits had similar fossil fuel flux estimates from the ODIAC product compared with the reference inventory product, but the ODIAC product had emissions that were more widespread across the domain away from the CBD. This led to aggregated estimates that were larger under the ODIAC inversion than the reference inversion. Compared to the reference, the ODIAC inversion attempted to reduce the aggregated flux for most months – and to a greater degree – to better match the observations, indicating that compared with the reference inventory, the ODIAC prior was most likely overestimating the amount of fossil fuel emissions from Cape Town to a greater extent for most parts of the study period."

P38, L15: "The inversion attempted to reduce the aggregated flux" means when the model tries to match the observations?

Response: In order to better match the observations, the inversion needed to reduce the fossil fuel fluxes implied by the ODIAC product, leading to a reduced aggregated flux over the domain. See above response.

P38, L18-20: Please provide estimates (in numbers) for both in the text so that the reader can clearly see the likely true emission estimates. Each inversion should have a uncertainty bound and then I don't understand what it means by "a much narrower uncertainty region than for either inversions." Response: This statement has been modified as follows, and the figure of the time series for this set of sensitivity tests has been updated to illustrate this idea and what the likely flux is: "When the two prior information products provide divergent prior flux estimates, such that the inversion reduced the flux for one product but increased the flux for the other, it suggests that the true flux lies somewhere between the posterior flux estimates from these two inversions. When the posterior aggregated flux was made smaller than the ODIAC prior but larger than the reference prior aggregated flux, such as during February and March 2013, the true aggregated flux should lie between these two posterior estimates. When the posterior flux was made smaller than the prior for both inversions, we could deduce that the true aggregated flux must be below the minimum of these two posterior estimates, and if we have accurate uncertainty estimates, the true flux should be no smaller than the lower uncertainty limit. Making use of the posterior uncertainties and the direction away from the prior in which the inversions made corrections, a region is suggested where the true flux is most likely to lie (Figure 9). For the CT domain, the inversion results suggest that over the spatial domain investigated, the flux is close to carbon neutral for the majority of the year."

P38, L26-28: 1 hour is too short. It should be useful to see the results based on 6 hours or 24 hours. I expect the length scale would be hours or even a couple of days.

Response: An additional case is added with correlation length of 7 hours. With a correlation length of 1 hour, the non-zero error correlations persist for observations at least 7 hours apart. We felt that there certainly should be error correlations, and therefore did not want to ignore these temporal

correlations, as is done for most of the urban inversions to date, but we also did not want make these correlations too long so that correlations would persist beyond a day, at least for the reference inversion.

P39, L17: This is not correct. Prior is just prior. Your sampling from a prior distribution with a fixed mean and a fixed covariance is still a priori info. It does not require the prior sample to be accurate. Response: The sentence in reference here is: "The posterior uncertainties reflect the reduction in uncertainty achieved by the inversion given that the prior uncertainties are accurate." What we meant here is that the inversion requires appropriate uncertainty limits in order to have the freedom to correct the prior fluxes such that the uncertainty limits around the posterior flux include the true flux. If the uncertainty limits are incorrectly specified such that they are too narrow, the inversion will still correct the flux in the right direction, but the uncertainty limits may not include the true flux. The way this paragraph is written in the original manuscript may be creating some confusion. Two issues are important here: 1.) The prior mean estimate. The inversion should always nudge the posterior mean closer to the true value. 2.) The uncertainty bounds placed around the prior mean estimate. The inversion will always result in a posterior uncertainty that is smaller than or equal to the prior uncertainty, even if the prior uncertainty is ridiculously small. In terms of the inversion's ability to push the posterior solution closer to the truth, this is determined by the prior uncertainty. Ideally, one would like to be able to set the prior uncertainty just large enough to allow the inversion to still be able to achieve a posterior solution close to the truth. The trick, of course, is getting the right uncertainty estimate.

P39, L19 - 20: This is because your data points are too small compared to the number of parameters to be solved. In other words, your inversion system is more dependent on the prior rather than observations. In this case, the posterior estimate for the individual pixels won't have much constraint; only the regional total emission may be estimated more or less independently, in the best case. From the Bayesian perspective, the only thing you can do is to report what your assumption was, what model was used and what the result is.

Response: The sentence referred to is: "It can be shown that in the absence of observation error, doubling or halving the prior uncertainty in the fluxes results in a respective doubling or halving of the posterior uncertainty." We agree that the observations only weakly constrain the fluxes. This is going the be the case for most urban inversions. There are few cities which have the luxury of being well constrained by observations. And that it is why it so important to get the uncertainty covariance parameters correct, particularly uncertainty correlation lengths, as these expand the influence of the observations onto surface pixels that may not be viewed directly by the observation network.

P39, L23 - 27: Not a Bayesian way of thinking, subject to criticism from frequentists. Response: The paragraph in question here is "This set of sensitivity tests demonstrated that if we wish to ensure that the uncertainty bounds around the posterior fluxes are within a prespecified margin, say 10% of the aggregated flux estimate, then we have to ensure that prior uncertainty that we begin with is sufficiently small. Assuming no large shifts in the mean estimate, it can be shown that if we wish to obtain an uncertainty estimate that is within 10% of the aggregated flux estimate, and we are able to reduce the uncertainty by 25% through the inversion, then the prior uncertainty estimate would need to be within 13.3% of the prior aggregated flux estimate."

We disagree that this is not a Bayesian way of thinking. In a Bayesian setting, we take advantage of the information we have to reduce the problem space to a narrower region. Normally when we assess a Bayesian inversion framework, we consider how much uncertainty reduction can the observations provide. The other side of the Bayesian solution is the prior information. We are considering by how much can we reduce the uncertainty of the posterior solution by ensuring that the prior information we start with in the inversion has sufficiently reduced the problem space.

For this methodology to be useful in the policy setting, the posterior estimates obtained from the inversion should ideally 1.) contain the true flux estimates, and 2.) the uncertainty limits should be narrow enough to determine if mitigation efforts are reducing emissions to a desired level with sufficient confidence. Since a great deal of resources already goes into the information used to provide prior flux estimates, the typical "expert-estimate" based approach of deciding on the uncertainty limits may never be good enough. Therefore alternative methods of determining the uncertainty parameters, such as the ML method mentioned in the next comment, or the Hierarchical Bayesian approach proposed, may be the best route forward.

P42, L14 - 15: Since the authors are using an analytical solutions for a Gaussian Likelihood function, they could use a simple maximum likelihood estimator for the length scale. Response: Michalak et al. (2005) and Wu et al. (2013) provides an ML approach for estimating the correlation length and other covariance parameters in an inversion. For a single inversion this requires an iterative method, such as the Gauss-Newton method, to derive these covariance parameters, even when uncertainty covariance matrix is assumed to be diagonal. That would not be feasible for this inversion frame-work, as the number of unknows is much larger, and we have not assumed a constant uncertainty for all sources, or assumed a single uncertainty scaling factor.

P42, L25 - 29: Please correct the sentences. Also, I don't know what the authors are trying to say here, except for the fact that a hierarchical approach may be better. Response: The point we are trying to make here is that the approach used for historical global and mesoscale inversions, whereby uncertainty covariance terms and uncertainty correlation lengths are driven by expert opinion, may not be feasible for a high resolution city-scale inversion due to sensitivity of the solution on these estimates. Instead, robust, data-driven estimates of these terms should be considered, such as the ML method described by Michalak et al (2005) or a Hierarchal Bayesian approach described by Ganesan et al (2009). This has not so far been done for city-scale high resolution inversions due to computational constraints. We showed that running weekly inversions solving for an average weekly flux gave a very similar solution to running a monthly inversion solving for average weekly fluxes. Computational costs could therefore be reduced by running shorter inversions, which is more feasibly for the ML or Hierarchical Bayesian approach requiring iterations of the inversion.

**Reviewer 2**

**Nickless et al review**

This manuscript describes a sensitivity study of an inversion of CO2 fluxes in and around Cape Town based on measurements at 3 sites. Cape Town is a city with a strong influence from biogenic fluxes and so provides a good case study for separating the anthropogenic influence from the biogenic influence. The main results from the inversion were published in a previous paper (Nickless et al., 2018). This manuscript concentrates on sensitivity studies on various aspects of the inversion, including the priors used for the biogenic and anthropogenic fluxes and the period over which inversions are averaged. This type of sensitivity analysis is undoubtedly important since cities emit such a large fraction of the global CO2, and there is a need to have robust and well understood inversion methodologies.

The paper is however hard to read. This is partly because it is pretty technical material and partly because so much information is included. This makes it difficult for an interested reader, let alone a casual one, to extract the main points, even after a careful reading. I do not get a feel for the main results from reading the abstract and do not think that the introduction sets the scene for the rest of the paper. I should note that the current discussion and conclusions do a better job of this.

Response: We have rewritten the Introduction and Methods sections in response to comments from the Editor and Reviewer. The introduction now contains a light introduction to Bayesian inversion studies in the context of city-scale inversions, and gives more discussion on the original Cape Town inversion study. We give a clearer explanation of why these sensitivity analyses were performed. The methodology section contains more of the details from the original paper, although we have kept this as lean as we can to avoid repeating too much of what is already described in Nickless et al 2018.

Overall, I think the manuscript could be publishable but only after major revision. I am not making many detailed suggestions as I think a considerable amount of work is needed and the first reviewer has made extensive and well thought out comments. My main comments are as follows:

1. The authors should clarify what the main findings are and then decide what material is needed to back that up in the introduction and in the main body of the manuscript. This should provide a firm basis on which to give a good understanding of the uncertainties and the implications described in the conclusions. That should result in a much tighter and probably shorter manuscript whose contents can be reflected in a clear abstract.

Response: Agreed. The introduction has been rewritten with this in mind.

2. In deciding what the main points are, the authors should consider whether ACP or GMD is the more appropriate home for the work. The ACP description includes the statement "The journal scope is focused on studies with general implications for atmospheric science rather than investigations that are primarily of local or technical interest." GMD "is an international scientific journal dedicated to the publication and public discussion of the description, development, and evaluation of numerical models of the Earth system and its components." Models include "geoscientific model descriptions, from statistical models to box models to GCMs." Response: Having read through the remit of GMD, I don't believe the type of sensitivity tests we have performed falls into the subject matter that is normally covered by this journal. If I was making changes to the atmospheric transport model it may be appropriate, but I think these types of statistical aspects of the inversion fit better into ACP. Previous studies on sensitivity analyses for cityscale inversions (focusing in this case on the observations used and the atmospheric transport model) have been published in ACP by Staufer et al. 2016 and on different priors used for a mesoscale inversion by Lauvaux et al. 2012. We have also made sure that the results and discussion now also emphasize what information the sensitivity tests provide about the flux of CO2 from this region.

3. I think that moving to GMD would allow the manuscript to be completely focussed on the technical aspects and might well make it easier to prepare.

Response: We have reduced the amount of technical detail in the manuscript and focussed more on the science and how these sensitivity tests inform future inversions.

4. The supplementary material largely consists of a series of plots which I am not sure are helpful, though I could be persuaded. I would think that some of the current paper could be put into a revised and reduced supplementary material.

Response: The purpose of the plots and tables in the supplementary material was to provide a type of look-up table so that if anyone were interested in a particular sensitivity test case, they could inspect exactly what the solution of the inversion looked like under these conditions, particularly for those cases which are only discussed briefly in the main text because the inversion solution was not sensitive to that particular change. This also avoids any issues related to selective reporting.

5. The present tense should be used for all the new results presented here, and the past tense should be used for previous work and much of the description of the measurements.I am not sure if I am typical, but the mixed use of tense misled me on a few occasions.Response: We have corrected the tense in the manuscript. Thank you for this guidance.

6. Some comment should be made about the important differences are present in the emissions products in sections 2.2 and 2.3. As it stands, it is hard to know what to keep in mind for later in the manuscript.

Response: More details have been added on the difference between the reference emission product and the alternative products, as described in the response to the first reviewer.

7. It would help to have a short summary of the results from Nickless et al (2018) at the start of Section 3.

Response: We have included in the new results section of the manuscript a brief summary of the results from the original Cape Town inversion.

8. Can percentages be included in the discussion of the changes vs the reference case? Response: We have included percentages when discussing the difference between the reference and alternative cases, at least when related to the change in the uncertainty. Reporting percentage changes with the total flux is difficult because the solution swings between being positive and negative for different inversions.

9. The aspect ratio in Figs 3, 4, and 9 should be increased. They are hard to read at the moment. Response: The figures have been replotted to be clearer and to focus only on the important aspects. The number of figures in the main manuscript has been reduced.

[revised manuscript text omitted]

$$J(\boldsymbol{s}) = \frac{1}{2} \left( (\boldsymbol{c}_{mod} - \boldsymbol{c})^T \mathbf{C}_{\boldsymbol{c}}^{-1} (\boldsymbol{c}_{mod} - \boldsymbol{c}) + (\boldsymbol{s} - \boldsymbol{s}_0)^T \mathbf{C}_{\boldsymbol{s}_0}^{-1} (\boldsymbol{s} - \mathbf{s}_0) \right)$$

8

(2)

and the solution for the posterior error covariance matrix for the sources,  $C_s$ ,

$$\underline{\mathbf{C}_{s}} \underline{=} \underbrace{\left(\mathbf{H}^{T} \mathbf{C}_{c}^{-1} \mathbf{H} + \mathbf{C}_{s_{0}}^{-1}\right)^{-1}}$$

$$\equiv \underline{\mathbf{C}_{s_0} - \mathbf{C}_{s_0} \mathbf{H}^T \left( \mathbf{H} \mathbf{C}_{s_0} \mathbf{H}^T + \mathbf{C}_c \right)^{-1} \mathbf{H} \mathbf{C}_{s_0}}$$

- 5 where *c* is the vector of concentration measurements from Robben Island and Hangklip measurement sites,  $s_0$  where *s* is the control vector of unknown surface fluxes and boundary concentrations we wish to solve for,  $s_0$  is the vector of prior estimates of these sources,  $C_c$  the error flux and boundary concentration estimates,  $C_c$  is the uncertainty covariance matrix of *c*, and  $C_{s_0}$  the prior the observations, and  $C_{s_0}$  is the uncertainty covariance matrix of  $s_0$ . H is the Jacobian matrix representing the first derivative of the modelled concentration,  $c_{mod}$ , at the observational site and dated with respect to the elements of *s*. H
- 10 projects the elements of *s* into the observation space of *c*the fluxes and boundary concentrations (Tarantola, 2005).

Minimising this cost function leads to the following solution:

 $c_{mod} = \mathbf{H}s.$

15
$$s = s_0 + C_{s_0} \mathbf{H}^T \left( \mathbf{H} C_{s_0} \mathbf{H}^T + \mathbf{C}_c \right)^{-1} (c - \mathbf{H} s_0)$$
(3)

The sources, s, consisted of gridded surface fluxes contained within the domain and concentrations of at the boundary. The spatial resolution of inversion was set at 1 by 1 and the extent of the domain was between 34.5° and 33.5° south and between 20 18.2° and 19.2° east. with posterior covariance matrix:

$$\mathbf{C}_{\mathbf{s}} = \left( \mathbf{H}^T \mathbf{C}_{\boldsymbol{c}}^{-1} \mathbf{H} + \mathbf{C}_{\boldsymbol{s}_0}^{-1} \right)^{-1}$$
(4)

[revised manuscript text omitted]
}\mathbf{s}_0 - \mathbf{c})^T (\mathbf{H}\mathbf{C}_{\mathbf{s}_0}\mathbf{H}^T + \mathbf{C}_{\mathbf{c}})^{-1} (\mathbf{H}\mathbf{s}_0 - \mathbf{c})$$
(13)

with degrees of freedom equal to  $\nu$ , the dimension of the data space – in this case the length of observations in the inversion. The squared residuals from the inversion (squared differences between observed and modelled concentrations) should follow the  $\chi^2$  distribution with degrees of freedom equal to the number of observations (Michalak et al., 2005; Tarantola, 2005). The expected value of  $\chi^2/\nu$  is one. Values lower than one indicate that the uncertainty is too large, and values greater than one

15 indicate that the uncertainty prescribed is lower than it should be. The error in the assignment of the uncertainty could be in either  $C_c$  or  $C_{s_0}$  (or both). In order to ensure the suitability of  $C_{s_0}$ , the prior uncertainty variances were multiplied by a factor of two. This ensured that the  $\chi^2/\nu$  statistic was close to a value of one for almost all months of the inversion. These details are provided in Nickless et al. (2018). Due to the length of time it takes to run a single inversion, we did not calculate an individual scaling parameter for each month.

**20 2.2 Alternative biogenic flux productSensitivity Tests**

**2.2.1 Alternative biogenic flux product**

As part of a project which aimed to assess assessing the carbon sinks of South Africa (DEA, 2015), a report together with monthly  $1 \text{ km} \times 1 \text{ km}$  estimates of terrestrial carbon stocks and fluxes were produced (Scholes et al., 2013). To estimate these fluxes, a distinction was made between carbon stocks in natural to semi-natural areas and those on transformed land, such

25 as annually-cropped cultivated land, plantation forests, and urban areas (which was based on the IPCC 2006 value for closed urban forests). We used these estimates of As a sensitivity test, the NEE and NPP in place of those from CABLE (inversion Carbon Assess from CABLE estimates used for the biogenic flux priors and their uncertainties were replaced with NEE and NPP from the carbon assessment product and the inversion rerun with these priors (inversion S1).

To estimate gross primary productivity (GPP), ten years (2001 to 2010) of monthly climatologies (temperature, rainfall, rel-30 ative humidity) and satellite products for photosynthetically active radiation (PAR) and fraction of absorbed photosynthetically active radiation (FAPAR) were assimilated. Autotrophic respiration (Ra) was calculated based on the inputs for temperature, above-ground biomass, below-ground biomass and FAPAR. NPP could then be calculated as NPP = GPP - Ra. The hetrotrophic component (Rh) of Ecosystem respiration (Re) was based on estimates of soil organic carbon stocks and above-ground litter. The basic calculation to obtain NEE was NEE = GPP - Re, and additional losses of  $CO_2$  through biomass burning, and export and import fluxes from harvest and trade-related activities were accounted for.

- To disaggregate the monthly products into day and night fluxes, it was assumed that all GPP took place during the day, and that half of Re occurred during the day and half at night. Therefore the weekly NEE and NPP estimates used for the prior information in the inversion were based on the GPP and respiration products from the assessment. The carbon assessment estimated the GPP flux for the year in the fynbos biome was estimated to be 521 g CO2 m-1-2/- year-1 with a standard deviation of 492 g CO2 m-1-2/- year-1 across pixels with 1 km2 resolution. Therefore, as for the CABLE estimates used in the reference
- 10 inversion, we assign uncertainties to the prior NEE estimates equal to the NPP estimate. A map of the prior daytime NEE fluxes in May 2012 from the CABLE and carbon assessment products is provided in Figure 1.

The biogenic  $CO_2$  fluxes are more homogeneous across the domain in the carbon assessment product. This can be explained by the products used as inputs for the estimation of the carbon stock components, such as FAPAR, which would not be expected to differ considerably from pixel to pixel in this domain. CABLE predicts greater  $CO_2$  uptake. The average  $CO_2$  flux over the

15 course of the study period and across the domain, was -41 g CO2 m-2week-1 according to the carbon assessment and -172 g CO2 m-2week-1 according to CABLE. The true flux is likely to be highly variable but close to carbon neutral over a long period of time (several years).

---

## Author Response (AR2)

Dear Editors,

Thank you for the review of the paper.

The following changes to the paper were requested:

Comments to the Author:
9, 20 - '..within the range of..'
Response: The required change has been made to this sentence.

12, 25 - '.. error dependent on the..'
Response: The spelling mistake has been corrected

15 - could you have one more look at the panels in Fig 1 and think if you can make the point more clearly in them? Because the top and bottom rows have different scales, it is hard to make meaningful comparisons or to link to the numbers in the text. I am not sure if the problem arises from the figure or from the brief reference in the text.
Response: Additional information has been added to the caption for the figure and to the text of the manuscript. The scales for the prior estimates (which can range between positive and negative values) was kept the same for both products, and the scale for the uncertainty estimates (which are spread across a wider range and are positive only) were kept the same.

The additional text to the caption:

[revised manuscript text omitted]